# DIFFUSION BRIDGE IMPLICIT MODELS

**Kaiwen Zheng**[12*], **Guande He**[1*], **Jianfei Chen**[1], **Fan Bao**[2], **Jun Zhu**[†123]
[1]Dept. of Comp. Sci. & Tech., Institute for AI, BNRist Center
[1]Tsinghua-Bosch Joint ML Center, Tsinghua University, Beijing, China
[2]Shengshu Technology, Beijing    [3]Pazhou Lab (Huangpu), Guangzhou, China
`zkwthu@gmail.com; guande.he17@outlook.com;`
`fan.bao@shengshu.ai; {jianfeic, dcszj}@tsinghua.edu.cn`

## ABSTRACT

Denoising diffusion bridge models (DDBMs) are a powerful variant of diffusion models for interpolating between two arbitrary paired distributions given as endpoints. Despite their promising performance in tasks like image translation, DDBMs require a computationally intensive sampling process that involves the simulation of a (stochastic) differential equation through hundreds of network evaluations. In this work, we take the first step in fast sampling of DDBMs without extra training, motivated by the well-established recipes in diffusion models. We generalize DDBMs via a class of non-Markovian diffusion bridges defined on the discretized timesteps concerning sampling, which share the same marginal distributions and training objectives, give rise to generative processes ranging from stochastic to deterministic, and result in diffusion bridge implicit models (DBIMs). DBIMs are not only up to $25\times$ faster than the vanilla sampler of DDBMs but also induce a novel, simple, and insightful form of ordinary differential equation (ODE) which inspires high-order numerical solvers. Moreover, DBIMs maintain the generation diversity in a distinguished way, by using a booting noise in the initial sampling step, which enables faithful encoding, reconstruction, and semantic interpolation in image translation tasks. Code is available at `https://github.com/thu-ml/DiffusionBridge`.

## 1 INTRODUCTION

Diffusion models (Song et al., 2021c; Sohl-Dickstein et al., 2015; Ho et al., 2020) represent a family of powerful generative models, with high-quality generation ability, stable training, and scalability to high dimensions. They have consistently obtained state-of-the-art performance in various domains, including image synthesis (Dhariwal & Nichol, 2021; Karras et al., 2022), speech and video generation (Chen et al., 2021a; Ho et al., 2022), controllable image manipulation (Nichol et al., 2022; Ramesh et al., 2022; Rombach et al., 2022; Meng et al., 2022), density estimation (Song et al., 2021b; Kingma et al., 2021; Lu et al., 2022a; Zheng et al., 2023b) and inverse problem solving (Chung et al., 2022; Kawar et al., 2022). They also act as fundamental components of modern text-to-image (Rombach et al., 2022) and text-to-video (Gupta et al., 2023; Bao et al., 2024) synthesis systems, ushering in the era of AI-generated content.

However, diffusion models are not well-suited for solving tasks like image translation or restoration, where the transport between two arbitrary probability distributions is to be modeled given paired endpoints. Diffusion models are rooted in a stochastic process that gradually transforms between data and noise, and the prior distribution is typically restricted to the "non-informative" random Gaussian noises. Adapting diffusion models to scenarios where a more informative prior naturally exists, such as image translation/restoration, involves modifying the generation pipeline (Meng et al., 2022; Su et al., 2022) or adding extra guidance terms during sampling (Chung et al., 2022; Kawar et al., 2022). On the one hand, these approaches are task-agnostic at training and adaptable to multiple tasks at inference time. On the other hand, despite recent advances in accelerated inverse problem solving (Liu et al., 2023a; Pandey et al., 2024), they inevitably deliver either sub-par performance or slow and resource-intensive inference compared to training-based ones. Tailored diffusion

---

*Equal contribution;    †The corresponding author.

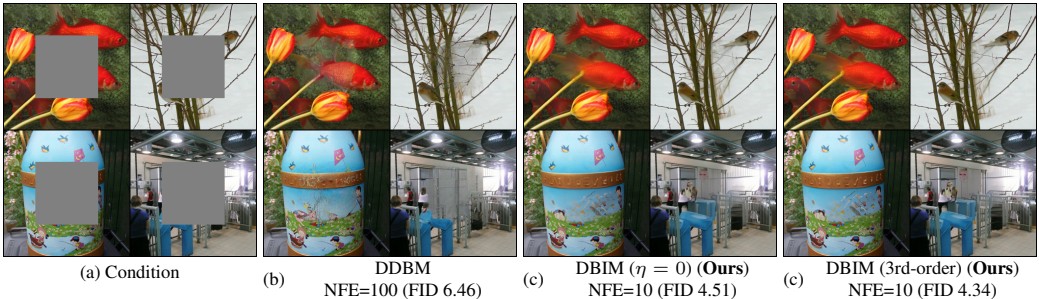

| (a) Condition | (b) | DDBM
NFE=100 (FID 6.46) | (c) | DBIM ($\eta = 0$) (**Ours**)
NFE=10 (FID 4.51) | (c) | DBIM (3rd-order) (**Ours**)
NFE=10 (FID 4.34) |

Figure 1: Inpainting results on the ImageNet 256×256 dataset (Deng et al., 2009) by DDBM (Zhou et al., 2023) with 100 number of function evaluations (NFE), and DBIM (ours) with only 10 NFE.

model variants become essential in task-specific scenarios where paired training data are available and fast inference is critical.

Recently, denoising diffusion bridge models (DDBMs) (Zhou et al., 2023) have emerged as a scalable and promising approach to solving the distribution translation tasks. By considering the reverse-time processes of a diffusion bridge, which represent diffusion processes conditioned on given endpoints, DDBMs offer a general framework for distribution translation. While excelling in image translation tasks with exceptional quality and fidelity, sampling from DDBMs requires simulating a (stochastic) differential equation corresponding to the reverse-time process. Even with the introduction of their hybrid sampler, achieving high-fidelity results for high-resolution images still demands over 100 steps. Compared to the efficient samplers for diffusion models (Song et al., 2021a; Zhang & Chen, 2022; Lu et al., 2022b), which require around 10 steps to generate reasonable samples, DDBMs are falling behind, urging the development of efficient variants.

This work represents the first pioneering effort toward accelerated sampling of DDBMs. As suggested by well-established recipes in diffusion models, training-free accelerations of diffusion sampling primarily focus on *reducing stochasticity* (e.g., the prominent denoising diffusion implicit models, DDIMs) and *utilizing higher-order information* (e.g., high-order solvers). We present diffusion bridge implicit models (DBIMs) as an approach that explores both aspects within the diffusion bridge framework. Firstly, we investigate the continuous-time forward process of DDBMs on discretized timesteps and generalize them to a series of non-Markovian diffusion bridges controlled by a variance parameter, while maintaining identical marginal distributions and training objectives as DDBMs. Secondly, the induced reverse generative processes correspond to sampling procedures of varying levels of stochasticity, including deterministic ones. Consequently, DBIMs can be viewed as a bridge counterpart and extension of DDIMs. Furthermore, in the continuous time limit, DBIMs can induce a novel form of ordinary differential equation (ODE), which is linked to the probability flow ODE (PF-ODE) in DDBMs while being simpler and significantly more efficient. The induced ODE also facilitates novel high-order numerical diffusion bridge solvers for faster convergence.

We demonstrate the superiority of DBIMs by applying them in image translation and restoration tasks, where they offer up to 25× faster sampling compared to DDBMs and achieve state-of-the-art performance on challenging high-resolution datasets. Unlike conventional diffusion sampling, the initial step in DBIMs is forced to be stochastic with a booting noise to avoid singularity issues arising from the fixed starting point on a bridge. By viewing the booting noise as the latent variable, DBIMs maintain the generation diversity of typical generative models while enabling faithful encoding, reconstruction, and semantically meaningful interpolation in the data space.

## 2 BACKGROUND

### 2.1 DIFFUSION MODELS

Given a $d$-dimensional data distribution $q_0(\boldsymbol{x}_0)$, diffusion models (Song et al., 2021c; Sohl-Dickstein et al., 2015; Ho et al., 2020) build a diffusion process by defining a forward stochastic differential equation (SDE) starting from $\boldsymbol{x}_0 \sim q_0$:

$$\mathrm{d}\boldsymbol{x}_t = f(t)\boldsymbol{x}_t \mathrm{d}t + g(t)\mathrm{d}\boldsymbol{w}_t \tag{1}$$

where $t \in [0, T]$ for some finite horizon $T$, $f, g : [0, T] \to \mathbb{R}$ is the scalar-valued drift and diffusion term, and $\boldsymbol{w}_t \in \mathbb{R}^d$ is a standard Wiener process. As a linear SDE, the forward process owns an analytic Gaussian transition kernel

$$q_{t|0}(\boldsymbol{x}_t|\boldsymbol{x}_0) = \mathcal{N}(\alpha_t \boldsymbol{x}_0, \sigma_t^2 \boldsymbol{I}) \tag{2}$$

by Itô's formula (Itô, 1951), where $\alpha_t, \sigma_t$ are called *noise schedules* satisfying $f(t) = \frac{\mathrm{d} \log \alpha_t}{\mathrm{d}t}$, $g^2(t) = \frac{\mathrm{d}\sigma_t^2}{\mathrm{d}t} - 2\frac{\mathrm{d}\log\alpha_t}{\mathrm{d}t}\sigma_t^2$ (Kingma et al., 2021). The forward SDE is accompanied by a series of marginal distributions $\{q_t\}_{t=0}^T$ of $\{\boldsymbol{x}_t\}_{t=0}^T$, and $f, g$ are properly designed so that the terminal distribution is approximately a pure Gaussian, i.e., $q_T(\boldsymbol{x}_T) \approx \mathcal{N}(\boldsymbol{0}, \sigma_T^2 \boldsymbol{I})$.

To sample from the data distribution $q_0(\boldsymbol{x}_0)$, we can solve the reverse SDE or probability flow ODE (Song et al., 2021c) from $t = T$ to $t = 0$:

$$\mathrm{d}\boldsymbol{x}_t = [f(t)\boldsymbol{x}_t - g^2(t)\nabla_{\boldsymbol{x}_t} \log q_t(\boldsymbol{x}_t)]\mathrm{d}t + g(t)\mathrm{d}\bar{\boldsymbol{w}}_t, \tag{3}$$

$$\mathrm{d}\boldsymbol{x}_t = \left[f(t)\boldsymbol{x}_t - \frac{1}{2}g^2(t)\nabla_{\boldsymbol{x}_t} \log q_t(\boldsymbol{x}_t)\right]\mathrm{d}t. \tag{4}$$

They share the same marginal distributions $\{q_t\}_{t=0}^T$ with the forward SDE, where $\bar{\boldsymbol{w}}_t$ is the reverse-time Wiener process, and the only unknown term $\nabla_{\boldsymbol{x}_t} \log q_t(\boldsymbol{x}_t)$ is the *score function* of the marginal density $q_t$. By denoising score matching (DSM) (Vincent, 2011), a score prediction network $\boldsymbol{s}_\theta(\boldsymbol{x}_t, t)$ can be parameterized to minimize $\mathbb{E}_t \mathbb{E}_{\boldsymbol{x}_0 \sim q_0(\boldsymbol{x}_0)} \mathbb{E}_{\boldsymbol{x}_t \sim q_{t|0}(\boldsymbol{x}_t|\boldsymbol{x}_0)} \left[w(t)\|\boldsymbol{s}_\theta(\boldsymbol{x}_t, t) - \nabla_{\boldsymbol{x}_t} \log q_{t|0}(\boldsymbol{x}_t|\boldsymbol{x}_0)\|_2^2\right]$, where $q_{t|0}$ is the analytic forward transition kernel and $w(t)$ is a positive weighting function. $\boldsymbol{s}_\theta$ can be plugged into the reverse SDE and the probability flow ODE to obtain the parameterized diffusion SDE and diffusion ODE. There are various dedicated solvers for diffusion SDE or ODE (Song et al., 2021a; Zhang & Chen, 2022; Lu et al., 2022b; Gonzalez et al., 2023).

## 2.2 Denoising Diffusion Bridge Models

Denoising diffusion bridge models (DDBMs) (Zhou et al., 2023) consider driving the diffusion process in Eqn. (1) to arrive at a particular point $\boldsymbol{y} \in \mathbb{R}^d$ almost surely via Doob's $h$-transform (Doob & Doob, 1984):

$$\mathrm{d}\boldsymbol{x}_t = f(t)\boldsymbol{x}_t\mathrm{d}t + g^2(t)\nabla_{\boldsymbol{x}_t} \log q(\boldsymbol{x}_T = \boldsymbol{y}|\boldsymbol{x}_t) + g(t)\mathrm{d}\boldsymbol{w}_t, \quad \boldsymbol{x}_0 \sim q_0 = p_{\text{data}}, \boldsymbol{x}_T = \boldsymbol{y}. \tag{5}$$

The endpoint $\boldsymbol{y}$ is not restricted to Gaussian noise as in diffusion models, but instead chosen as informative priors (such as the degraded image in image restoration tasks). Given a starting point $\boldsymbol{x}_0$, the process in Eqn. (5) also owns an analytic forward transition kernel

$$q(\boldsymbol{x}_t|\boldsymbol{x}_0, \boldsymbol{x}_T) = \mathcal{N}(a_t\boldsymbol{x}_T + b_t\boldsymbol{x}_0, c_t^2 \boldsymbol{I}), \quad a_t = \frac{\alpha_t}{\alpha_T}\frac{\text{SNR}_T}{\text{SNR}_t}, b_t = \alpha_t(1 - \frac{\text{SNR}_T}{\text{SNR}_t}), c_t^2 = \sigma_t^2(1 - \frac{\text{SNR}_T}{\text{SNR}_t}) \tag{6}$$

which forms a *diffusion bridge*, and $\text{SNR}_t = \alpha_t^2/\sigma_t^2$ is the signal-to-noise ratio at time $t$. DDBMs show that the forward process Eqn. (5) is associated with a reverse SDE and a probability flow ODE starting from $\boldsymbol{x}_T = \boldsymbol{y}$:

$$\mathrm{d}\boldsymbol{x}_t = \left[f(t)\boldsymbol{x}_t - g^2(t)\left(\nabla_{\boldsymbol{x}_t} \log q(\boldsymbol{x}_t|\boldsymbol{x}_T = \boldsymbol{y}) - \nabla_{\boldsymbol{x}_t} \log q_{T|t}(\boldsymbol{x}_T = \boldsymbol{y}|\boldsymbol{x}_t))\right)\right]\mathrm{d}t + g(t)\mathrm{d}\bar{\boldsymbol{w}}_t, \tag{7}$$

$$\mathrm{d}\boldsymbol{x}_t = \left[f(t)\boldsymbol{x}_t - g^2(t)\left(\frac{1}{2}\nabla_{\boldsymbol{x}_t} \log q(\boldsymbol{x}_t|\boldsymbol{x}_T = \boldsymbol{y}) - \nabla_{\boldsymbol{x}_t} \log q_{T|t}(\boldsymbol{x}_T = \boldsymbol{y}|\boldsymbol{x}_t)\right)\right]\mathrm{d}t. \tag{8}$$

They share the same marginal distributions $\{q(\boldsymbol{x}_t|\boldsymbol{x}_T = \boldsymbol{y})\}_{t=0}^T$ with the forward process, where $\bar{\boldsymbol{w}}_t$ is the reverse-time Wiener process, $q_{T|t}$ is analytically known similar to Eqn. (2), and the only unknown term $\nabla_{\boldsymbol{x}_t} \log q(\boldsymbol{x}_t|\boldsymbol{x}_T = \boldsymbol{y})$ is the *bridge score function*. Denoising bridge score matching (DBSM) is proposed to learn the unknown score term $q(\boldsymbol{x}_t|\boldsymbol{x}_T = \boldsymbol{y})$ with a parameterized network $\boldsymbol{s}_\theta(\boldsymbol{x}_t, t, \boldsymbol{y})$, by minimizing

$$\mathcal{L}_w(\theta) = \mathbb{E}_t \mathbb{E}_{(\boldsymbol{x}_0, \boldsymbol{y}) \sim p_{\text{data}}(\boldsymbol{x}_0, \boldsymbol{y})} \mathbb{E}_{\boldsymbol{x}_t \sim q(\boldsymbol{x}_t|\boldsymbol{x}_0, \boldsymbol{x}_T = \boldsymbol{y})} \left[w(t)\|\boldsymbol{s}_\theta(\boldsymbol{x}_t, t, \boldsymbol{y}) - \nabla_{\boldsymbol{x}_t} \log q(\boldsymbol{x}_t|\boldsymbol{x}_0, \boldsymbol{x}_T = \boldsymbol{y})\|_2^2\right] \tag{9}$$

where $q(\boldsymbol{x}_t|\boldsymbol{x}_0, \boldsymbol{x}_T = \boldsymbol{y})$ is the forward transition kernel in Eqn. (6) and $w(t)$ is a positive weighting function. To sample from diffusion bridges with Eqn. (7) and Eqn. (8), DDBMs propose a high-order hybrid sampler that alternately simulates the ODE and SDE steps to enhance the sample quality, inspired by the Heun sampler in diffusion models (Karras et al., 2022). However, it is not dedicated to diffusion bridges and lacks theoretical insights in developing efficient diffusion samplers.

## 3 GENERATIVE MODEL THROUGH NON-MARKOVIAN DIFFUSION BRIDGES

We start by examining the forward process of the diffusion bridge (Eqn. (5)) on a set of discretized timesteps $0 = t_0 < t_1 < \cdots < t_{N-1} < t_N = T$ that will be used for reverse sampling. Since the bridge score $\nabla_{\boldsymbol{x}_t} \log q(\boldsymbol{x}_t|\boldsymbol{x}_T)$ only depends on the marginal distribution $q(\boldsymbol{x}_t|\boldsymbol{x}_T)$, we can construct alternative probabilistic models that induce new sampling procedures while reusing the learned bridge score $\boldsymbol{s}_\theta(\boldsymbol{x}_t, t, \boldsymbol{x}_T)$, as long as they agree on the $N$ marginals $\{q(\boldsymbol{x}_{t_n}|\boldsymbol{x}_T)\}_{n=0}^{N-1}$.

### 3.1 NON-MARKOVIAN DIFFUSION BRIDGES AS FORWARD PROCESS

We consider a family of probability distributions $q^{(\rho)}(\boldsymbol{x}_{t_{0:N-1}}|\boldsymbol{x}_T)$, controlled by a variance parameter $\rho \in \mathbb{R}^{N-1}$:

$$q^{(\rho)}(\boldsymbol{x}_{t_{0:N-1}}|\boldsymbol{x}_T) = q_0(\boldsymbol{x}_{t_0}) \prod_{n=1}^{N-1} q^{(\rho)}(\boldsymbol{x}_{t_n}|\boldsymbol{x}_0, \boldsymbol{x}_{t_{n+1}}, \boldsymbol{x}_T) \tag{10}$$

where $q_0$ is the data distribution at time 0 and for $1 \leq n \leq N-1$

$$q^{(\rho)}(\boldsymbol{x}_{t_n}|\boldsymbol{x}_0, \boldsymbol{x}_{t_{n+1}}, \boldsymbol{x}_T) = \mathcal{N}(a_{t_n}\boldsymbol{x}_T + b_{t_n}\boldsymbol{x}_0 + \sqrt{c_{t_n}^2 - \rho_n^2} \frac{\boldsymbol{x}_{t_{n+1}} - a_{t_{n+1}}\boldsymbol{x}_T - b_{t_{n+1}}\boldsymbol{x}_0}{c_{t_{n+1}}}, \rho_n^2 \boldsymbol{I}) \tag{11}$$

where $\rho_n$ is the $n$-th element of $\rho$ satisfying $\rho_{N-1} = c_{t_{N-1}}$, and $a_t, b_t, c_t$ are terms related to the noise schedule, as defined in the original diffusion bridge (Eqn. (6)). Intuitively, this decreases the variance (noise level) of the bridge while incorporating additional noise components from the last step. Under this construction, we can prove that $q^{(\rho)}$ maintains consistency in marginal distributions with the original forward process $q$ governed by Eqn. (5).

**Proposition 3.1** (Marginal Preservation, proof in Appendix B.1). *For $0 \leq n \leq N-1$, we have* $q^{(\rho)}(\boldsymbol{x}_{t_n}|\boldsymbol{x}_T) = q(\boldsymbol{x}_{t_n}|\boldsymbol{x}_T)$.

The definition of $q^{(\rho)}$ in Eqn. (10) represents the inference process, since it is factorized as the distribution of $\boldsymbol{x}_{t_n}$ given $\boldsymbol{x}_{t_{n+1}}$ at the previous timestep. Conversely, the forward process $q^{(\rho)}(\boldsymbol{x}_{t_{n+1}}|\boldsymbol{x}_0, \boldsymbol{x}_{t_n}, \boldsymbol{x}_T)$ can be induced by Bayes' rule (Appendix C.1). As $\boldsymbol{x}_{t_{n+1}}$ in $q^{(\rho)}$ can simultaneously depend on $\boldsymbol{x}_{t_n}$ and $\boldsymbol{x}_0$, we refer to it as *non-Markovian diffusion bridges*, in contrast to Markovian ones (such as Brownian bridges, and the diffusion bridge defined by the forward SDE in Eqn. (5)) which should satisfy $q(\boldsymbol{x}_{t_{n+1}}|\boldsymbol{x}_0, \boldsymbol{x}_{t_n}, \boldsymbol{x}_T) = q(\boldsymbol{x}_{t_{n+1}}|\boldsymbol{x}_{t_n}, \boldsymbol{x}_T)$.

### 3.2 REVERSE GENERATIVE PROCESS AND EQUIVALENT TRAINING OBJECTIVE

Eqn. (10) can be naturally transformed into a parameterized and learnable generative model, by replacing the unknown $\boldsymbol{x}_0$ in Eqn. (10) with a *data predictor* $\boldsymbol{x}_\theta(\boldsymbol{x}_t, t, \boldsymbol{x}_T)$. Intuitively, $\boldsymbol{x}_t$ on the diffusion bridge is a weighted mixture of $\boldsymbol{x}_T, \boldsymbol{x}_0$ and some random Gaussian noise according to Eqn. (6), where the weightings $a_t, b_t, c_t$ are determined by the timestep $t$. The network $\boldsymbol{x}_\theta$ is trained to recover the clean data $\boldsymbol{x}_0$ given $\boldsymbol{x}_t, \boldsymbol{x}_T$ and $t$.

Specifically, we define the generative process starting from $\boldsymbol{x}_T$ as

$$p_\theta(\boldsymbol{x}_{t_n}|\boldsymbol{x}_{t_{n+1}}, \boldsymbol{x}_T) = \begin{cases} \mathcal{N}(\boldsymbol{x}_\theta(\boldsymbol{x}_{t_1}, t_1, \boldsymbol{x}_T), \rho_0^2 \boldsymbol{I}), & n = 0 \\ q^{(\rho)}(\boldsymbol{x}_{t_n}|\boldsymbol{x}_\theta(\boldsymbol{x}_{t_{n+1}}, t_{n+1}, \boldsymbol{x}_T), \boldsymbol{x}_{t_{n+1}}, \boldsymbol{x}_T), & 1 \leq n \leq N-1 \end{cases} \tag{12}$$

and the joint distribution as $p_\theta(\boldsymbol{x}_{t_{0:N-1}}|\boldsymbol{x}_T) = \prod_{n=0}^{N-1} p_\theta(\boldsymbol{x}_{t_n}|\boldsymbol{x}_{t_{n+1}}, \boldsymbol{x}_T)$. To optimize the network parameter $\theta$, we can adopt the common variational inference objective as in DDPMs (Ho et al., 2020), except that the distributions are conditioned on $\boldsymbol{x}_T$:

$$\mathcal{J}^{(\rho)}(\theta) = \mathbb{E}_{q(\boldsymbol{x}_T)} \mathbb{E}_{q^{(\rho)}(\boldsymbol{x}_{t_{0:N-1}}|\boldsymbol{x}_T)} \left[ \log q^{(\rho)}(\boldsymbol{x}_{t_{1:N-1}}|\boldsymbol{x}_0, \boldsymbol{x}_T) - \log p_\theta(\boldsymbol{x}_{t_{0:N-1}}|\boldsymbol{x}_T) \right] \tag{13}$$

It seems that the DDBM objective $\mathcal{L}_w$ in Eqn. (9) is distinct from $\mathcal{J}^{(\rho)}$: respectively, they are defined on continuous and discrete timesteps; they originate from score matching and variational inference; they have different parameterizations of score and data prediction[1]. However, we show they are equivalent by focusing on the discretized timesteps and transforming the parameterization.

---

[1] The diffusion bridge models are usually parameterized differently from score prediction, but can be converted to score prediction. See Appendix F.1 for details.

Table 1: Comparison between different diffusion models and diffusion bridge models.

| | Diffusion Models | | | Diffusion Bridge Models | | |
| | DDPM (Ho et al., 2020) | ScoreSDE (Song et al., 2021c) | DDIM (Song et al., 2021a) | I²SB (Liu et al., 2023b) | DDBM (Zhou et al., 2023) | DBIM (**Ours**) |
|---|---|---|---|---|---|---|
| Noise Schedule | VP | Any | Any | VE | Any | Any |
| Timesteps | Discrete | Continuous | Discrete | Discrete | Continuous | Discrete |
| Forward Distribution | $q(\boldsymbol{x}_n\|\boldsymbol{x}_0)$ | $q(\boldsymbol{x}_t\|\boldsymbol{x}_0)$ | $q(\boldsymbol{x}_n\|\boldsymbol{x}_{n-1},\boldsymbol{x}_0)$ | $q(\boldsymbol{x}_n\|\boldsymbol{x}_0,\boldsymbol{x}_N)$ | $q(\boldsymbol{x}_t\|\boldsymbol{x}_0,\boldsymbol{x}_T)$ | $q(\boldsymbol{x}_{t_{n+1}}\|\boldsymbol{x}_0,\boldsymbol{x}_{t_n},\boldsymbol{x}_T)$ |
| Inference Process | $p_\theta(\boldsymbol{x}_{n-1}\|\boldsymbol{x}_n)$ | SDE/ODE | $p_\theta(\boldsymbol{x}_{n-1}\|\boldsymbol{x}_n)$ | $p_\theta(\boldsymbol{x}_{n-1}\|\boldsymbol{x}_n)$ | SDE/ODE | $p_\theta(\boldsymbol{x}_{t_n}\|\boldsymbol{x}_{t_{n+1}},\boldsymbol{x}_T)$ |
| Non-Markovian | ✗ | ✗ | ✓ | ✗ | ✗ | ✓ |

**Proposition 3.2** (Training Equivalence, proof in Appendix B.2). *For $\rho > 0$, there exists certain weights $\gamma$ so that $\mathcal{J}^{(\rho)}(\theta) = \mathcal{L}_\gamma(\theta) + C$ on the discretized timesteps $\{t_n\}_{n=1}^N$, where $C$ is a constant irrelevant to $\theta$. Besides, the bridge score predictor $\boldsymbol{s}_\theta$ in $\mathcal{L}_\gamma(\theta)$ has the following relationship with the data predictor $\boldsymbol{x}_\theta$ in $\mathcal{J}^{(\rho)}(\theta)$:*

$$\boldsymbol{s}_\theta(\boldsymbol{x}_t, t, \boldsymbol{x}_T) = -\frac{\boldsymbol{x}_t - a_t\boldsymbol{x}_T - b_t\boldsymbol{x}_\theta(\boldsymbol{x}_t, t, \boldsymbol{x}_T)}{c_t^2} \tag{14}$$

Though the weighting $\gamma$ may not precisely match the actual weighting $w$ for training $\boldsymbol{s}_\theta$, this discrepancy doesn't affect our utilization of $\boldsymbol{s}_\theta$ (Appendix C.2). Hence, it is reasonable to reuse the network trained by $\mathcal{L}$ while leveraging various $\rho$ for improved sampling efficiency.

## 4 SAMPLING WITH GENERALIZED DIFFUSION BRIDGES

Now that we have confirmed the rationality and built the theoretical foundations for applying the generalized diffusion bridge $p_\theta$ to pretrained DDBMs, a range of inference processes is now at our disposal, controlled by the variance parameter $\rho$. This positions us to explore the resultant sampling procedures and the effects of $\rho$ in pursuit of better and more efficient generation.

### 4.1 DIFFUSION BRIDGE IMPLICIT MODELS

Suppose we sample in reverse time on the discretized timesteps $0 = t_0 < t_1 < \cdots < t_{N-1} < t_N = T$. The number $N$ and the schedule of sampling steps can be made independently of the original timesteps on which the bridge model is trained, whether discrete (Liu et al., 2023b) or continuous (Zhou et al., 2023). According to the generative process of $p_\theta$ in Eqn. (12), the updating rule from $t_{n+1}$ to $t_n$ is described by

$$\boldsymbol{x}_{t_n} = a_{t_n}\boldsymbol{x}_T + b_{t_n}\hat{\boldsymbol{x}}_0 + \sqrt{c_{t_n}^2 - \rho_n^2}\underbrace{\frac{\boldsymbol{x}_{t_{n+1}} - a_{t_{n+1}}\boldsymbol{x}_T - b_{t_{n+1}}\hat{\boldsymbol{x}}_0}{c_{t_{n+1}}}}_{\text{predicted noise } \hat{\boldsymbol{\epsilon}}} + \rho_n\boldsymbol{\epsilon}, \quad \boldsymbol{\epsilon} \sim \mathcal{N}(\boldsymbol{0}, \boldsymbol{I}) \tag{15}$$

where $\hat{\boldsymbol{x}}_0 = \boldsymbol{x}_\theta(\boldsymbol{x}_{t_{n+1}}, t_{n+1}, \boldsymbol{x}_T)$ denotes the predicted clean data at time 0.

**Intuition of the Sampling Procedure**   Intuitively, the form of Eqn. (15) resembles the forward transition kernel of the diffusion bridge in Eqn. (6) (which can be rewritten as $\boldsymbol{x}_t = a_t\boldsymbol{x}_T + b_t\boldsymbol{x}_0 + c_t\boldsymbol{\epsilon}, \boldsymbol{\epsilon} \sim \mathcal{N}(\boldsymbol{0}, \boldsymbol{I})$). In comparison, $\boldsymbol{x}_0$ is substituted with the predicted $\hat{\boldsymbol{x}}_0$, and a portion of the standard Gaussian noise $\boldsymbol{\epsilon}$ now stems from the predicted noise $\hat{\boldsymbol{\epsilon}}$. The predicted noise $\hat{\boldsymbol{\epsilon}}$ is derived from $\boldsymbol{x}_{t_{n+1}}$ at the previous timestep and can be expressed by the predicted clean data $\hat{\boldsymbol{x}}_0$.

**Effects of the Variance Parameter**   We investigate the effects of the variance parameter $\rho$ from the theoretical perspective by considering two extreme cases. Firstly, we note that when $\rho_n = \sigma_{t_n}\sqrt{1 - \frac{\text{SNR}_{t_{n+1}}}{\text{SNR}_{t_n}}}$ for each $0 \leq n \leq N - 1$, the $\boldsymbol{x}_T$ term in Eqn. (15) is canceled out. In this scenario, the forward process in Eqn. (4.1) becomes a Markovian bridge (see details in Appendix C.1). Besides, the inference process will get rid of $\boldsymbol{x}_T$ and simplify to $p_\theta(\boldsymbol{x}_{t_n}|\boldsymbol{x}_{t_{n+1}})$, akin to the sampling mechanism in DDPMs (Ho et al., 2020). Secondly, when $\rho_n = 0$ for each $0 \leq n \leq N-1$, the inference process will be free from random noise and composed of deterministic iterative updates, characteristic of an implicit probabilistic model (Mohamed & Lakshminarayanan,

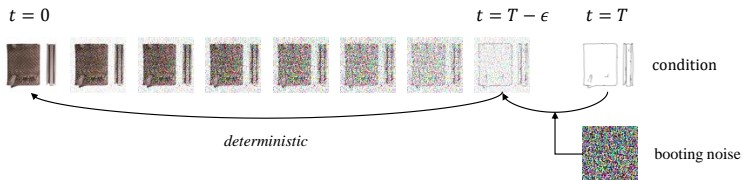

$t = 0$ $t = T - \epsilon$ $t = T$

condition

*deterministic*

booting noise

Figure 2: Illustration of the DBIM's deterministic sampling procedure when $\rho = 0$.

2016). Consequently, we name the resulting model *diffusion bridge implicit models* (DBIMs), drawing parallels with denoising diffusion implicit models (DDIMs) (Song et al., 2021a). DBIMs serve as the bridge counterpart and extension of DDIMs, as illustrated in Table 1.

When we choose $\rho$ that lies between these two boundary cases, we can obtain non-Markovian diffusion bridges with intermediate and non-zero stochastic levels. Such bridges may potentially yield superior sample quality. We present detailed ablations in Section 6.1.

**The Singularity at the Initial Step for Deterministic Sampling**  One important aspect to note regarding DBIMs is that its initial step exhibits singularity when $\rho = 0$, a property essentially distinct from DDIMs in diffusion models. Specifically, in the initial step we have $t_{n+1} = T$, and $c_{t_{n+1}}$ in the denominator in Eqn. (15) equals 0. This phenomenon can be understood intuitively: given a fixed starting point $\boldsymbol{x}_T$, the variable $\boldsymbol{x}_t$ for $t < T$ is typically still stochastically distributed (the marginal $p_\theta(\boldsymbol{x}_t|\boldsymbol{x}_T)$ is not a Dirac distribution). For instance, in inpainting tasks, there should be various plausible complete images corresponding to a fixed masked image. However, a fully deterministic sampling procedure disrupts such stochasticity.

To be theoretically robust, we employ the other boundary choice $\rho_n = \sigma_{t_n} \sqrt{1 - \frac{\text{SNR}_{t_{n+1}}}{\text{SNR}_{t_n}}}$ in the initial step[2], which is aligned with our previous restriction that $\rho_{N-1} = c_{t_{N-1}}$. This will introduce an additional standard Gaussian noise $\epsilon$ which we term as the *booting noise*. It accounts for the stochasticity of the final sample $\boldsymbol{x}_0$ under a given fixed $\boldsymbol{x}_T$ and can be viewed as the latent variable. We illustrate the complete DBIM pipeline in Figure 2.

### 4.2 CONNECTION TO PROBABILITY FLOW ODE

It is intuitive to perceive that the deterministic sampling can be related to solving an ODE. By setting $\rho = 0, t_{n+1} = t$ and $t_{n+1} - t_n = \Delta t$ in Eqn. (15), the DBIM updating rule can be reorganized as $\frac{\boldsymbol{x}_{t-\Delta t}}{c_{t-\Delta t}} = \frac{\boldsymbol{x}_t}{c_t} + \left( \frac{a_{t-\Delta t}}{c_{t-\Delta t}} - \frac{a_t}{c_t} \right) \boldsymbol{x}_T + \left( \frac{b_{t-\Delta t}}{c_{t-\Delta t}} - \frac{b_t}{c_t} \right) \boldsymbol{x}_\theta(\boldsymbol{x}_t, t, \boldsymbol{x}_T)$. As $a_t, b_t, c_t$ are continuous functions of time $t$ defined in Eqn. (6), the ratios $\frac{a_t}{c_t}$ and $\frac{b_t}{c_t}$ also remain continuous functions of $t$. Therefore, DBIM ($\rho = 0$) can be treated as an Euler discretization of the following ordinary differential equation (ODE):

$$\mathrm{d}\left( \frac{\boldsymbol{x}_t}{c_t} \right) = \boldsymbol{x}_T \mathrm{d}\left( \frac{a_t}{c_t} \right) + \boldsymbol{x}_\theta(\boldsymbol{x}_t, t, \boldsymbol{x}_T) \mathrm{d}\left( \frac{b_t}{c_t} \right) \tag{16}$$

Though it does not resemble a conventional ODE involving $\mathrm{d}t$, the two infinitesimal terms $\mathrm{d}\left( \frac{a_t}{c_t} \right)$ and $\mathrm{d}\left( \frac{b_t}{c_t} \right)$ can be expressed with $\mathrm{d}t$ by the chain rule of derivatives. The ODE form also suggests that with a sufficient number of discretization steps, we can reverse the sampling process and obtain encodings of the observed data, which can be useful for interpolation or other downstream tasks.

In DDBMs, the PF-ODE (Eqn. (8)) involving $\mathrm{d}\boldsymbol{x}_t$ and $\mathrm{d}t$ is proposed and used for deterministic sampling. We reveal in the following proposition that our ODE in Eqn. (16) can exactly yield the PF-ODE without relying on the advanced Kolmogorov forward (or Fokker-Planck) equation.

**Proposition 4.1** (Equivalence to Probability Flow ODE, proof in Appendix B.3). *Suppose $\boldsymbol{s}_\theta(\boldsymbol{x}_t, t, \boldsymbol{x}_T)$ is learned as the ground-truth bridge score $\nabla_{\boldsymbol{x}_t} \log q(\boldsymbol{x}_t|\boldsymbol{x}_T)$, and $\boldsymbol{x}_\theta$ is related to $\boldsymbol{s}_\theta$ through Eqn. (14), then Eqn. (16) can be converted to the PF-ODE (Eqn. (8)) proposed in DDBMs.*

---

[2]With this choice, at the initial step $n = N-1$, we have $\rho_n = \sigma_{t_n} \sqrt{1 - \frac{\text{SNR}_{t_T}}{\text{SNR}_{t_n}}} = c_{t_n} \Rightarrow \sqrt{c_{t_n}^2 - \rho_n^2} = 0$, so $c_{t_{n+1}}$ in the denominator in Eqn. (15) will be canceled out.

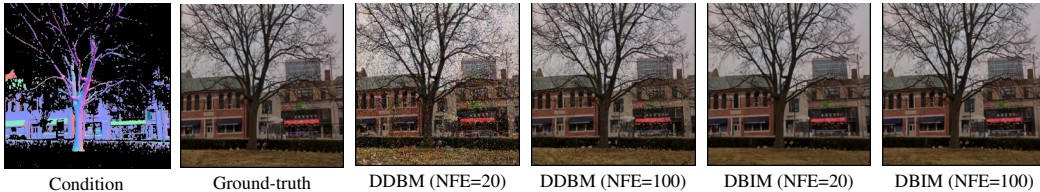

| Condition | Ground-truth | DDBM (NFE=20) | DDBM (NFE=100) | DBIM (NFE=20) | DBIM (NFE=100) |

Figure 3: Image translation results on the DIODE-Outdoor dataset with DDBM and DBIM.

Though the conversion from our ODE to the PF-ODE is straightforward, the reverse conversion can be non-trivial and require complex tools such as exponential integrators (Calvo & Palencia, 2006; Hochbruck et al., 2009) (Appendix C.4). We highlight our differences from the PF-ODE in DDBMs: (1) Our ODE has a novel form with exceptional neatness. (2) Despite their theoretical equivalence, our ODE describes the evolution of $\frac{x_t}{c_t}$ rather than $x_t$, and its discretization is performed with respect to $\mathrm{d}\left(\frac{a_t}{c_t}\right)$ and $\mathrm{d}\left(\frac{b_t}{c_t}\right)$ instead of $\mathrm{d}t$. (3) Empirically, DBIMs ($\rho = 0$) prove significantly more efficient than the Euler discretization of the PF-ODE, thereby accelerating DDBMs by a substantial margin. (4) In contrast to the fully deterministic ODE, DBIMs are capable of various stochastic levels to achieve the best generation quality under the same sampling steps.

### 4.3 EXTENSION TO HIGH-ORDER METHODS

The simplicity and efficiency of our ODE (Eqn. (16)) also inspire novel high-order numerical solvers tailored for DDBMs, potentially bringing faster convergence than the first-order Euler discretization. Specifically, using the time change-of-variable $\lambda_t = \log\left(\frac{b_t}{c_t}\right) = \frac{1}{2}(\mathrm{SNR}_t - \mathrm{SNR}_T)$, the solution of Eqn. (16) from time $t$ to time $s < t$ can be represented as

$$\boldsymbol{x}_s = \frac{c_s}{c_t}\boldsymbol{x}_t + \left(a_s - \frac{c_s}{c_t}a_t\right)\boldsymbol{x}_T + c_s \int_{\lambda_t}^{\lambda_s} e^\lambda \boldsymbol{x}_\theta(\boldsymbol{x}_{t_\lambda}, t_\lambda, \boldsymbol{x}_T)\mathrm{d}\lambda \tag{17}$$

where $t_\lambda$ is the inverse function of $\lambda_t$. The intractable integral can be approximated by Taylor expansion of $\boldsymbol{x}_\theta$ and finite difference estimations of high-order derivatives, following well-established numerical methods (Hochbruck & Ostermann, 2005) and their extensive application in diffusion models (Zhang & Chen, 2022; Lu et al., 2022b; Gonzalez et al., 2023). We present the derivations of our high-order solvers in Appendix D, and the detailed algorithm in Appendix E.

## 5 RELATED WORK

We present detailed related work in Appendix A, including diffusion models, diffusion bridge models, and fast sampling techniques. We additionally discuss some special cases of DBIM and their connection to flow matching, DDIM and posterior sampling in Appendix C.3.

## 6 EXPERIMENTS

In this section, we show that DBIMs surpass the original sampling procedure of DDBMs by a large margin, in terms of both sample quality and sample efficiency. We also showcase DBIM's capabilities in latent-space encoding, reconstruction, and interpolation using deterministic sampling. All comparisons between DBIMs and DDBMs are conducted using identically trained models. For DDBMs, we employ their proposed hybrid sampler for sampling. For DBIMs, we control the variance parameter $\rho$ by interpolating between its boundary selections:

$$\rho_n = \eta \sigma_{t_n} \sqrt{1 - \frac{\mathrm{SNR}_{t_{n+1}}}{\mathrm{SNR}_{t_n}}}, \quad \eta \in [0, 1] \tag{18}$$

where $\eta = 0$ and $\eta = 1$ correspond to deterministic sampling and Markovian stochastic sampling.

We conduct experiments including (1) image-to-image translation tasks on Edges→Handbags (Isola et al., 2017) ($64 \times 64$) and DIODE-Outdoor (Vasiljevic et al., 2019) ($256 \times 256$) (2) image restoration task of inpainting on ImageNet (Deng et al., 2009) ($256 \times 256$) with $128 \times 128$ center mask.

Table 2: Quantitative results in the image translation task. [†]Baseline results are taken directly from DDBMs, where they did not report the exact NFE. Gray-colored rows denote methods that do not require paired training but only a prior diffusion model trained on the target domain.

| | | Edges→Handbags ($64 \times 64$) | | | | DIODE-Outdoor ($256 \times 256$) | | | |
|---|---|---|---|---|---|---|---|---|---|
| | NFE | FID ↓ | IS ↑ | LPIPS ↓ | MSE ↓ | FID ↓ | IS ↑ | LPIPS ↓ | MSE ↓ |
| DDIB (Su et al., 2022) | $\geq 40$[†] | 186.84 | 2.04 | 0.869 | 1.05 | 242.3 | 4.22 | 0.798 | 0.794 |
| SDEdit (Meng et al., 2022) | $\geq 40$ | 26.5 | **3.58** | 0.271 | 0.510 | 31.14 | 5.70 | 0.714 | 0.534 |
| Pix2Pix (Isola et al., 2017) | 1 | 74.8 | 3.24 | 0.356 | 0.209 | 82.4 | 4.22 | 0.556 | 0.133 |
| I²SB (Liu et al., 2023b) | $\geq 40$ | 7.43 | 3.40 | 0.244 | 0.191 | 9.34 | 5.77 | 0.373 | 0.145 |
| DDBM (Zhou et al., 2023) | 118 | 1.83 | 3.73 | 0.142 | 0.040 | 4.43 | **6.21** | 0.244 | 0.084 |
| DDBM (Zhou et al., 2023) | 200 | **0.88** | 3.69 | 0.110 | 0.006 | 3.34 | 5.95 | 0.215 | 0.020 |
| DBIM (Ours) | 20 | 1.74 | 3.63 | **0.095** | **0.005** | 4.99 | 6.10 | 0.201 | **0.017** |
| DBIM (Ours) | 100 | **0.89** | 3.62 | 0.100 | 0.006 | **2.57** | 6.06 | **0.198** | 0.018 |

Table 3: Quantative results in the image restoration task.

| Inpainting | | ImageNet ($256 \times 256$) | |
|---|---|---|---|
| *Center* ($128 \times 128$) | NFE | FID ↓ | CA ↑ |
| DDRM (Kawar et al., 2022) | 20 | 24.4 | 62.1 |
| ΠGDM (Song et al., 2023a) | 100 | 7.3 | 72.6 |
| DDNM (Wang et al., 2023) | 100 | 15.1 | 55.9 |
| Palette (Saharia et al., 2022) | 1000 | 6.1 | 63.0 |
| I²SB (Liu et al., 2023b) | 10 | 5.24 | 66.1 |
| I²SB (Liu et al., 2023b) | 20 | 4.98 | 65.9 |
| I²SB (Liu et al., 2023b) | 1000 | 4.9 | 66.1 |
| DDBM (Zhou et al., 2023) | 500 | 4.27 | 71.8 |
| DBIM (Ours) | 10 | 4.48 | 71.3 |
| DBIM (Ours) | 20 | 4.07 | 72.3 |
| DBIM (Ours) | 100 | **3.88** | **72.7** |

Table 4: Ablation of the variance parameter controlled by $\eta$ for image restoration, measured by FID.

| Sampler | | NFE | | | | | | |
|---|---|---|---|---|---|---|---|---|
| | | 5 | 10 | 20 | 50 | 100 | 200 | 500 |
| | | Inpainting, ImageNet ($256 \times 256$), *Center* ($128 \times 128$) | | | | | | |
| $\eta$ | 0.0 | **6.08** | 4.51 | 4.11 | 3.95 | 3.91 | 3.91 | 3.91 |
| | 0.3 | 6.12 | **4.48** | 4.09 | 3.95 | 3.92 | 3.90 | 3.88 |
| | 0.5 | 6.25 | 4.52 | **4.07** | 3.92 | 3.90 | 3.84 | 3.86 |
| | 0.8 | 6.81 | 4.79 | 4.16 | **3.91** | **3.88** | 3.84 | 3.81 |
| | 1.0 | 8.62 | 5.61 | 4.51 | 4.05 | 3.91 | **3.80** | **3.80** |
| DDBM | | 275.25 | 57.18 | 29.65 | 10.63 | 6.46 | 4.95 | 4.27 |

We report the Fréchet inception distance (FID) (Heusel et al., 2017) for all experiments, and additionally measure Inception Scores (IS) (Barratt & Sharma, 2018), Learned Perceptual Image Patch Similarity (LPIPS) (Zhang et al., 2018), Mean Square Error (MSE) (for image-to-image translation) and Classifier Accuracy (CA) (for image inpainting), following previous works (Liu et al., 2023b; Zhou et al., 2023). The metrics are computed using the complete training set for Edges→Handbags and DIODE-Outdoor, and 10k images from validation set for ImageNet. We provide the inference time comparison in Appendix G.1. Additional experiment details are provided in Appendix F.

## 6.1 SAMPLE QUALITY AND EFFICIENCY

We present the quantitative results of DBIMs in Table 2 and Table 3, compared with baselines including GAN-based, diffusion-based and bridge-based methods[3]. We set the number of function evaluations (NFEs) of DBIM to 20 and 100 to demonstrate both efficiency at small NFEs and quality at large NFEs. We select $\eta$ from the set [0.0, 0.3, 0.5, 0.8, 1.0] for DBIM and report the best results.

In image translation tasks, DDBM achieves the best sample quality (measured by FID) among the baselines, but requires NFE > 100. In contrast, DBIM with only NFE = 20 already surpasses all baselines, performing better than or on par with DDBM at NFE = 118. When increasing the NFE to 100, DBIM further improves the sample quality and outperforms DDBM with NFE = 200 on DIODE-Outdoor. In the more challenging image inpainting task on ImageNet $256 \times 256$, the superiority of DBIM is highlighted even further. In particular, DBIM with NFE = 20 outperforms all baselines, including DDBM with NFE = 500, achieving a $25\times$ speed-up. With NFE = 100, DBIM continues to improve sample quality, reaching a FID lower than 4 for the first time.

The comparison of visual quality is illustrated in Figure 1 and Figure 3, where DBIM produces smoother outputs with significantly fewer noisy artifacts compared to DDBM's hybrid sampler. Additional samples are provided in Appendix H.

**Ablation of the Variance Parameter** We investigate the impact of the variance parameter $\rho$ (controlled by $\eta$) to identify how the level of stochasticity affects sample quality across various NFEs, as shown in Table 4 and Table 5. For image translation tasks, we consistently observe that employing

---

[3]It is worth noting that, the released checkpoints of I²SB are actually **flow matching/interpolant** models instead of bridge models, as they (1) start with noisy conditions instead of clean conditions and (2) perform a straight interpolation between the condition and the sample without adding extra intermediate noise.

Table 5: Ablation of the variance parameter controlled by $\eta$ for image translation, measured by FID.

| Sampler | | NFE | | | | | | | | | | | | |
|---|---|---|---|---|---|---|---|---|---|---|---|---|---|---|
| | | 5 | 10 | 20 | 50 | 100 | 200 | 500 | 5 | 10 | 20 | 50 | 100 | 200 | 500 |
| | | Image Translation, Edges→Handbags ($64 \times 64$) | | | | | | | Image Translation, DIODE-Outdoor ($256 \times 256$) | | | | | | |
| | 0.0 | **3.62** | **2.49** | **1.76** | **1.17** | **0.91** | **0.75** | 0.65 | **14.25** | **7.96** | **4.97** | **3.18** | **2.56** | **2.26** | **2.10** |
| | 0.3 | 3.64 | 2.53 | 1.81 | 1.21 | 0.94 | 0.76 | 0.65 | 14.48 | 8.25 | 5.22 | 3.37 | 2.68 | 2.33 | 2.12 |
| $\eta$ | 0.5 | 3.69 | 2.61 | 1.91 | 1.30 | 1.00 | 0.81 | 0.67 | 14.93 | 8.75 | 5.68 | 3.71 | 2.92 | 2.47 | 2.17 |
| | 0.8 | 3.87 | 2.91 | 2.25 | 1.58 | 1.23 | 0.96 | 0.76 | 16.41 | 10.30 | 6.98 | 4.63 | 3.58 | 2.90 | 2.41 |
| | 1.0 | 4.21 | 3.38 | 2.72 | 1.96 | 1.50 | 1.15 | 0.85 | 19.17 | 12.59 | 8.85 | 5.98 | 4.55 | 3.59 | 2.82 |
| DDBM | | 317.22 | 137.15 | 46.74 | 7.79 | 2.40 | 0.88 | **0.53** | 328.33 | 151.93 | 41.03 | 15.19 | 6.54 | 3.34 | 2.26 |

a deterministic sampler with $\eta = 0$ yields superior performance compared to stochastic samplers with $\eta > 0$. We attribute it to the characteristics of the datasets, where the target image is highly correlated with and dependent on the condition, resulting in a generative model that lacks diversity. In this case, a straightforward mapping without the involvement of stochasticity is preferred. Conversely, for image inpainting on the more diverse dataset ImageNet $256 \times 256$, the parameter $\eta$ exhibits significance across different NFEs. When NFE $\leq 20$, $\eta = 0$ is near the optimal choice, with FID steadily increasing as $\eta$ ascends. However, when NFE $\geq 50$, a relatively large level of stochasticity at $\eta = 0.8$ or even $\eta = 1$ yields optimal FID. Notably, the FID of $\eta = 0$ converges to 3.91 at NFE = 100, with no further improvement at larger NFEs, indicating convergence to the ground-truth sample by the corresponding PF-ODE. This observation aligns with diffusion models, where deterministic sampling facilitates rapid convergence, while introducing stochasticity in sampling enhances diversity, ultimately culminating in the highest sample quality when NFE is substantial.

Table 6: The effects of high-order methods, measured by FID.

| Sampler | NFE | | | | | | | | | | | | | | |
|---|---|---|---|---|---|---|---|---|---|---|---|---|---|---|---|
| | 5 | 10 | 20 | 50 | 100 | 5 | 10 | 20 | 50 | 100 | 5 | 10 | 20 | 50 | 100 |
| | Image Translation | | | | | | | | | | Inpainting | | | | |
| | Edges→Handbags ($64 \times 64$) | | | | | DIODE-Outdoor ($256 \times 256$) | | | | | ImageNet ($256 \times 256$) | | | | |
| DBIM ($\eta = 0$) | 3.62 | 2.49 | 1.76 | 1.17 | 0.91 | 14.25 | 7.96 | 4.97 | 3.18 | 2.56 | 6.08 | 4.51 | 4.11 | 3.95 | **3.91** |
| DBIM (2nd-order) | 3.44 | 2.16 | 1.48 | 0.99 | **0.79** | 13.54 | 7.18 | 4.34 | 2.87 | 2.41 | 5.53 | **4.33** | **4.07** | 3.94 | **3.91** |
| DBIM (3rd-order) | **3.40** | **2.12** | **1.45** | 0.97 | **0.79** | **13.41** | **7.01** | **4.20** | **2.84** | **2.40** | **5.50** | 4.34 | **4.07** | 3.93 | **3.91** |

**High-Order Methods**   We further demonstrate the effects of high-order methods by comparing them to deterministic DBIM, the first-order case. As shown in Table 6, high-order methods consistently improve FID scores in image translation tasks, as well as in inpainting tasks when NFE$\leq 50$, resulting in enhanced generation quality in the low NFE regime. Besides, the 3rd-order variant performs slightly better than the 2nd-order variant. However, in contrast to the numerical solvers in diffusion models, the benefits of high-order extensions are relatively minor in diffusion bridges and less pronounced than the improvement when adjusting $\eta$ from 1 to 0. Nevertheless, high-order DBIMs are significantly more efficient than DDBM's PF-ODE-based high-order solvers.

As illustrated in Figure 1, our high-order sampler produces images of similar semantic content to the first-order case, using the same booting noise. In contrast, the visual quality is improved with finer textures, resulting in better FID. This indicates that the high-order gradient information from past network outputs benefits the generation quality by adding high-frequency visual details.

**Generation Diversity**   We quantitatively measure the generation diversity by the diversity score, calculated as the pixel-level variance of multiple generations, following CMDE (Batzolis et al., 2021) and BBDM (Li et al., 2023). As detailed in Appendix G.2, increasing NFE or decreasing $\eta$ can both increase the diversity score, confirming the effect of the booting noise.

## 6.2   RECONSTRUCTION AND INTERPOLATION

As discussed in Section 4.2, the deterministic nature of DBIMs at $\eta = 0$ and its connection to neural ODEs enable faithful encoding and reconstruction by treating the booting noise as the latent variable. Furthermore, employing spherical linear interpolation in the latent space and subsequently decoding

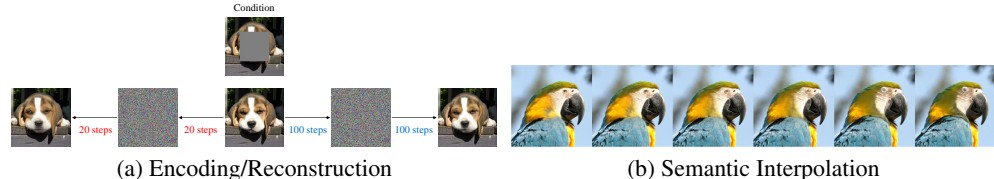

(a) Encoding/Reconstruction                              (b) Semantic Interpolation

Figure 4: Illustration of generation diversity with deterministic DBIMs.

back to the image space allows for semantic image interpolation in image translation and image restoration tasks. These capabilities cannot be achieved by DBIMs with $\eta > 0$, or by DDBM's hybrid sampler which incorporates stochastic steps. We showcase the encoding and decoding results in Figure 4a, indicating that accurate reconstruction is achievable with a sufficient number of sampling steps. We also illustrate the interpolation process in Figure 4b.

# 7    CONCLUSION

In this work, we introduce diffusion bridge implicit models (DBIMs) for accelerated sampling of DDBMs without extra training. In contrast to DDBM's continuous-time generation processes, we concentrate on discretized sampling steps and propose a series of generalized diffusion bridge models including non-Markovian variants. The induced sampling procedures serve as bridge counterparts and extensions of DDIMs and are further extended to develop high-order numerical solvers, filling the missing perspectives in the context of diffusion bridges. Experiments on high-resolution datasets and challenging inpainting tasks demonstrate DBIM's superiority in both the sample quality and sample efficiency, achieving state-of-the-art FID scores with 100 steps and providing up to $25\times$ acceleration of DDBM's sampling procedure.

**Limitations and Failure Cases**  Despite the notable speed-up for diffusion bridge models, DBIMs still lag behind GAN-based methods in one-step generation. The generation quality is unsatisfactory when NFE is small, and blurry regions still exist even using high-order methods (Figure 1). This is not fast enough for real-time applications. Besides, as a training-free inference algorithm, DBIM cannot surpass the capability and quality upper bound of the pretrained diffusion bridge model. In difficult and delicate inpainting scenarios, such as human faces and hands, DBIM fails to fix the artifacts, even under large NFEs.

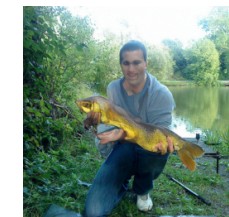

Figure 5: DBIM case ($\eta = 0$, NFE=500).

# ACKNOWLEDGMENTS

This work was supported by the NSFC Projects (Nos. 62350080, 62106120, 92270001), the National Key Research and Development Program of China (No. 2021ZD0110502), Tsinghua Institute for Guo Qiang, and the High Performance Computing Center, Tsinghua University. J.Z was also supported by the XPlorer Prize.

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

## A    RELATED WORK

**Fast Sampling of Diffusion Models**    Fast sampling of diffusion models can be classified into training-free and training-based methods. A prevalent training-free fast sampler is the denoising diffusion implicit models (DDIMs) (Song et al., 2021a) that employ alternative non-Markovian generation processes in place of DDPMs, a discrete-time diffusion model. ScoreSDE (Song et al., 2021c) further links discrete-time DDPMs to continuous-time score-based models, unrevealing the generation process to be ordinary and stochastic differential equations (ODEs and SDEs). DDIM can be generalized to develop integrators for broader diffusion models (Zhang et al., 2022; Pandey et al., 2023). The concept of implicit sampling, in a broad sense, can also be extended to discrete diffusion models (Chen et al., 2024; Zheng et al., 2024), although there are fundamental differences in their underlying mechanisms. Subsequent training-free samplers concentrate on developing dedicated numerical solvers to the diffusion ODE or SDE, particularly Heun's methods (Karras et al., 2022) and exponential integrators (Zhang & Chen, 2022; Lu et al., 2022b; Zheng et al., 2023a; Gonzalez et al., 2023). These methods typically require around 10 steps for high-quality generation. In contrast, training-based methods, particularly adversarial distillation (Sauer et al., 2023) and consistency distillation (Song et al., 2023b; Kim et al., 2023), become notable for their ability to achieve high-quality generation with just one or two steps. Our work serves as a thorough exploration of training-free fast sampling of DDBMs. Exploring bridge distillation methods, such as consistency bridge distillation (He et al., 2024), would be promising future research avenues to decrease the inference cost further. Infrastructure improvements, such as quantized or sparse attention (Zhang et al., 2025a;b; 2024), can also be used to accelerate the inference of diffusion bridge models.

**Diffusion Bridges**    Diffusion bridges (De Bortoli et al., 2021; Chen et al., 2021b; Liu et al., 2023b; Somnath et al., 2023; Zhou et al., 2023; Chen et al., 2023; Shi et al., 2024; Deng et al., 2024) are a promising generative variant of diffusion models for modeling the transport between two arbitrary distributions. One line of work is the diffusion Schrodinger bridge models (De Bortoli et al., 2021; Chen et al., 2021b; Shi et al., 2024; Deng et al., 2024), which solves an entropy-regularized optimal transport problem between two probability distributions. However, their reliance on expensive iterative procedures has limited their application scope, particularly for high-dimensional data. Subsequent works have endeavored to enhance the tractability of the Schrodinger bridge problem by making assumptions such as paired data (Liu et al., 2023b; Somnath et al., 2023; Chen et al., 2023). On the other hand, DDBMs (Zhou et al., 2023) construct diffusion bridges via Doob's $h$-transform, offering a reverse-time perspective of a diffusion process conditioned on given endpoints. This approach aligns the design spaces and training algorithms of DDBMs closely with those of score-based generative models, leading to state-of-the-art performance in image translation tasks. However, the sampling procedure of DDBMs still relies on inefficient simulations of differential equations, lacking theoretical insights to develop efficient samplers. BBDM (Li et al., 2023) and I³SB (Wang et al., 2024a) extend the concept of DDIM to the contexts of Brownian bridge and I²SB (Liu et al., 2023b), respectively. SinSR (Wang et al., 2024b) is also motivated by DDIM, while the application is concentrated on the mean-reverting diffusion process, which ends in a Gaussian instead of a delta distribution. In contrast to them, our work provides the first systematic exploration of implicit sampling within the broader DDBM framework, offering theoretical insights and connections while also proposing novel high-order diffusion bridge solvers.

## B    PROOFS

### B.1    PROOF OF PROPOSITION 3.1

*Proof.* Since $q^{(\rho)}$ in Eqn. (10) is factorized as $q^{(\rho)}(\boldsymbol{x}_{t_{0:N-1}}|\boldsymbol{x}_T) = q_0(\boldsymbol{x}_0)q^{(\rho)}(\boldsymbol{x}_{t_{1:N-1}}|\boldsymbol{x}_0, \boldsymbol{x}_T)$ where $q^{(\rho)}(\boldsymbol{x}_{t_{1:N-1}}|\boldsymbol{x}_0, \boldsymbol{x}_T) = \prod_{n=1}^{N-1} q^{(\rho)}(\boldsymbol{x}_{t_n}|\boldsymbol{x}_0, \boldsymbol{x}_{t_{n+1}}, \boldsymbol{x}_T)$, we have $q^{(\rho)}(\boldsymbol{x}_0|\boldsymbol{x}_T) = q_0(\boldsymbol{x}_0) = q(\boldsymbol{x}_0|\boldsymbol{x}_T)$, which proves the case for $n = 0$. For $1 \leq n \leq N - 1$, we have

$$q^{(\rho)}(\boldsymbol{x}_{t_n}|\boldsymbol{x}_T) = \int q^{(\rho)}(\boldsymbol{x}_{t_n}|\boldsymbol{x}_0, \boldsymbol{x}_T)q^{(\rho)}(\boldsymbol{x}_0|\boldsymbol{x}_T)\mathrm{d}\boldsymbol{x}_0 \tag{19}$$

and

$$q(\boldsymbol{x}_{t_n}|\boldsymbol{x}_T) = \int q(\boldsymbol{x}_{t_n}|\boldsymbol{x}_0, \boldsymbol{x}_T)q(\boldsymbol{x}_0|\boldsymbol{x}_T)\mathrm{d}\boldsymbol{x}_0 \tag{20}$$

Since $q^{(\rho)}(\boldsymbol{x}_0|\boldsymbol{x}_T) = q(\boldsymbol{x}_0|\boldsymbol{x}_T)$, we only need to prove $q^{(\rho)}(\boldsymbol{x}_{t_n}|\boldsymbol{x}_0, \boldsymbol{x}_T) = q(\boldsymbol{x}_{t_n}|\boldsymbol{x}_0, \boldsymbol{x}_T)$.

Firstly, when $n = N - 1$, we have $t_{n+1} = T$. Note that $\rho$ is restricted by $\rho_{N-1} = c_{t_{N-1}}$, and Eqn. (11) becomes

$$q^{(\rho)}(\boldsymbol{x}_{t_{N-1}}|\boldsymbol{x}_0, \boldsymbol{x}_T) = \mathcal{N}(a_{t_{N-1}}\boldsymbol{x}_T + b_{t_{N-1}}\boldsymbol{x}_0, c_{t_{N-1}}^2 \boldsymbol{I}) \tag{21}$$

which is exactly the same as the forward transition kernel of $q$ in Eqn. (6). Therefore, $q^{(\rho)}(\boldsymbol{x}_{t_n}|\boldsymbol{x}_0, \boldsymbol{x}_T) = q(\boldsymbol{x}_{t_n}|\boldsymbol{x}_0, \boldsymbol{x}_T)$ holds for $n = N - 1$.

Secondly, suppose $q^{(\rho)}(\boldsymbol{x}_{t_n}|\boldsymbol{x}_0, \boldsymbol{x}_T) = q(\boldsymbol{x}_{t_n}|\boldsymbol{x}_0, \boldsymbol{x}_T)$ holds for $n = k$, we aim to prove that it holds for $n = k - 1$. Specifically, $q^{(\rho)}(\boldsymbol{x}_{t_{k-1}}|\boldsymbol{x}_0, \boldsymbol{x}_T)$ can be expressed as

$$\begin{aligned}
q^{(\rho)}(\boldsymbol{x}_{t_{k-1}}|\boldsymbol{x}_0, \boldsymbol{x}_T) &= \int q^{(\rho)}(\boldsymbol{x}_{t_{k-1}}|\boldsymbol{x}_0, \boldsymbol{x}_{t_k}, \boldsymbol{x}_T) q^{(\rho)}(\boldsymbol{x}_{t_k}|\boldsymbol{x}_0, \boldsymbol{x}_T) \mathrm{d}\boldsymbol{x}_{t_k} \\
&= \int q^{(\rho)}(\boldsymbol{x}_{t_{k-1}}|\boldsymbol{x}_0, \boldsymbol{x}_{t_k}, \boldsymbol{x}_T) q(\boldsymbol{x}_{t_k}|\boldsymbol{x}_0, \boldsymbol{x}_T) \mathrm{d}\boldsymbol{x}_{t_k} \\
&= \int \mathcal{N}(\boldsymbol{x}_{t_{k-1}}; \boldsymbol{\mu}_{k-1|k}, \rho_{k-1}^2 \boldsymbol{I}) \mathcal{N}(\boldsymbol{x}_{t_k}; a_{t_k}\boldsymbol{x}_T + b_{t_k}\boldsymbol{x}_0, c_{t_k}^2 \boldsymbol{I}) \mathrm{d}\boldsymbol{x}_{t_k}
\end{aligned} \tag{22}$$

where

$$\boldsymbol{\mu}_{k-1|k} = a_{t_{k-1}}\boldsymbol{x}_T + b_{t_{k-1}}\boldsymbol{x}_0 + \sqrt{c_{t_{k-1}}^2 - \rho_{k-1}^2} \frac{\boldsymbol{x}_{t_k} - a_{t_k}\boldsymbol{x}_T - b_{t_k}\boldsymbol{x}_0}{c_{t_k}} \tag{23}$$

From (Bishop & Nasrabadi, 2006) (2.115), $q^{(\rho)}(\boldsymbol{x}_{t_{k-1}}|\boldsymbol{x}_0, \boldsymbol{x}_T)$ is a Gaussian, denoted as $\mathcal{N}(\boldsymbol{\mu}_{k-1}, \Sigma_{k-1})$, where

$$\begin{aligned}
\boldsymbol{\mu}_{k-1} &= a_{t_{k-1}}\boldsymbol{x}_T + b_{t_{k-1}}\boldsymbol{x}_0 + \sqrt{c_{t_{k-1}}^2 - \rho_{k-1}^2} \frac{a_{t_k}\boldsymbol{x}_T + b_{t_k}\boldsymbol{x}_0 - a_{t_k}\boldsymbol{x}_T - b_{t_k}\boldsymbol{x}_0}{c_{t_k}} \\
&= a_{t_{k-1}}\boldsymbol{x}_T + b_{t_{k-1}}\boldsymbol{x}_0
\end{aligned} \tag{24}$$

and

$$\begin{aligned}
\Sigma_{k-1} &= \rho_{k-1}^2 \boldsymbol{I} + \frac{\sqrt{c_{t_{k-1}}^2 - \rho_{k-1}^2}}{c_{t_k}} c_{t_k}^2 \frac{\sqrt{c_{t_{k-1}}^2 - \rho_{k-1}^2}}{c_{t_k}} \boldsymbol{I} \\
&= c_{t_{k-1}}^2 \boldsymbol{I}
\end{aligned} \tag{25}$$

Therefore, $q^{(\rho)}(\boldsymbol{x}_{t_{k-1}}|\boldsymbol{x}_0, \boldsymbol{x}_T) = q(\boldsymbol{x}_{t_{k-1}}|\boldsymbol{x}_0, \boldsymbol{x}_T) = \mathcal{N}(a_{t_{k-1}}\boldsymbol{x}_T + b_{t_{k-1}}\boldsymbol{x}_0, c_{t_{k-1}}^2 \boldsymbol{I})$. By mathematical deduction, $q^{(\rho)}(\boldsymbol{x}_{t_n}|\boldsymbol{x}_0, \boldsymbol{x}_T) = q(\boldsymbol{x}_{t_n}|\boldsymbol{x}_0, \boldsymbol{x}_T)$ holds for every $1 \leq n \leq N - 1$, which completes the proof. $\qquad\square$

### B.2 PROOF OF PROPOSITION 3.2

*Proof.* Substituting Eqn. (10) and the joint distribution into Eqn. (13), we have

$$\begin{aligned}
&\mathcal{J}^{(\rho)}(\theta) \\
&= \mathbb{E}_{q(\boldsymbol{x}_T)} \mathbb{E}_{q^{(\rho)}(\boldsymbol{x}_{t_{0:N-1}}|\boldsymbol{x}_T)} \left[ \log q^{(\rho)}(\boldsymbol{x}_{t_{1:N-1}}|\boldsymbol{x}_0, \boldsymbol{x}_T) - \log p_\theta(\boldsymbol{x}_{t_{0:N-1}}|\boldsymbol{x}_T) \right] \\
&= \mathbb{E}_{q(\boldsymbol{x}_T)} \mathbb{E}_{q^{(\rho)}(\boldsymbol{x}_{t_{0:N-1}}|\boldsymbol{x}_T)} \left[ \sum_{n=1}^{N-1} \log q^{(\rho)}(\boldsymbol{x}_{t_n}|\boldsymbol{x}_0, \boldsymbol{x}_{t_{n+1}}, \boldsymbol{x}_T) - \sum_{n=0}^{N-1} \log p_\theta(\boldsymbol{x}_{t_n}|\boldsymbol{x}_{t_{n+1}}, \boldsymbol{x}_T) \right] \\
&= \sum_{n=1}^{N-1} \mathbb{E}_{q(\boldsymbol{x}_T)} \mathbb{E}_{q^{(\rho)}(\boldsymbol{x}_0, \boldsymbol{x}_{t_{n+1}}|\boldsymbol{x}_T)} \left[ D_{\mathrm{KL}}(q^{(\rho)}(\boldsymbol{x}_{t_n}|\boldsymbol{x}_0, \boldsymbol{x}_{t_{n+1}}, \boldsymbol{x}_T) \| p_\theta(\boldsymbol{x}_{t_n}|\boldsymbol{x}_{t_{n+1}}, \boldsymbol{x}_T)) \right] \\
&\quad - \mathbb{E}_{q(\boldsymbol{x}_T)} \mathbb{E}_{q^{(\rho)}(\boldsymbol{x}_0, \boldsymbol{x}_{t_1}|\boldsymbol{x}_T)} \left[ \log p_\theta(\boldsymbol{x}_0|\boldsymbol{x}_{t_1}, \boldsymbol{x}_T) \right]
\end{aligned} \tag{26}$$

where

$$\begin{aligned}
&D_{\mathrm{KL}}(q^{(\rho)}(\boldsymbol{x}_{t_n}|\boldsymbol{x}_0, \boldsymbol{x}_{t_{n+1}}, \boldsymbol{x}_T) \| p_\theta(\boldsymbol{x}_{t_n}|\boldsymbol{x}_{t_{n+1}}, \boldsymbol{x}_T)) \\
&= D_{\mathrm{KL}}(q^{(\rho)}(\boldsymbol{x}_{t_n}|\boldsymbol{x}_0, \boldsymbol{x}_{t_{n+1}}, \boldsymbol{x}_T) \| q^{(\rho)}(\boldsymbol{x}_{t_n}|\boldsymbol{x}_\theta(\boldsymbol{x}_{t_{n+1}}, t_{n+1}, \boldsymbol{x}_T), \boldsymbol{x}_{t_{n+1}}, \boldsymbol{x}_T)) \\
&= \frac{d_n^2 \|\boldsymbol{x}_\theta(\boldsymbol{x}_{t_{n+1}}, t_{n+1}, \boldsymbol{x}_T) - \boldsymbol{x}_0\|_2^2}{2\rho_n^2}
\end{aligned} \tag{27}$$

where we have denoted $d_n := b_{t_n} - \sqrt{c_{t_n}^2 - \rho_n^2} \frac{b_{t_{n+1}}}{c_{t_{n+1}}}$. Besides, we have

$$
\begin{aligned}
\log p_\theta(\boldsymbol{x}_0|\boldsymbol{x}_{t_1}, \boldsymbol{x}_T) &= \log q^{(\rho)}(\boldsymbol{x}_0|\boldsymbol{x}_\theta(\boldsymbol{x}_{t_1}, t_1, \boldsymbol{x}_T), \boldsymbol{x}_{t_1}, \boldsymbol{x}_T) \\
&= \log \mathcal{N}(\boldsymbol{x}_\theta(\boldsymbol{x}_{t_1}, t_1, \boldsymbol{x}_T), \rho_0^2 \boldsymbol{I}) \\
&= -\frac{\|\boldsymbol{x}_\theta(\boldsymbol{x}_{t_1}, t_1, \boldsymbol{x}_T) - \boldsymbol{x}_0\|_2^2}{2\rho_0^2} + C
\end{aligned}
\tag{28}
$$

where $C$ is irrelevant to $\theta$. According to Eqn. (6), the conditional score is

$$
\nabla_{\boldsymbol{x}_t} \log q(\boldsymbol{x}_t|\boldsymbol{x}_0, \boldsymbol{x}_T) = -\frac{\boldsymbol{x}_t - a_t \boldsymbol{x}_T - b_t \boldsymbol{x}_0}{c_t^2}
\tag{29}
$$

Therefore,

$$
\begin{aligned}
&\|\boldsymbol{x}_\theta(\boldsymbol{x}_{t_n}, t_n, \boldsymbol{x}_T) - \boldsymbol{x}_0\|_2^2 \\
&= \frac{c_{t_n}^4}{b_{t_n}^2} \left\| -\frac{\boldsymbol{x}_{t_n} - a_{t_n} \boldsymbol{x}_T - b_{t_n} \boldsymbol{x}_\theta(\boldsymbol{x}_{t_n}, t_n, \boldsymbol{x}_T)}{c_{t_n}^2} - \left( -\frac{\boldsymbol{x}_{t_n} - a_{t_n} \boldsymbol{x}_T - b_{t_n} \boldsymbol{x}_0}{c_{t_n}^2} \right) \right\|_2^2 \\
&= \frac{c_{t_n}^4}{b_{t_n}^2} \|\boldsymbol{s}_\theta(\boldsymbol{x}_{t_n}, t_n, \boldsymbol{x}_T) - \nabla_{\boldsymbol{x}_{t_n}} \log q(\boldsymbol{x}_{t_n}|\boldsymbol{x}_0, \boldsymbol{x}_T)\|_2^2
\end{aligned}
\tag{30}
$$

where $\boldsymbol{s}_\theta$ is related to $\boldsymbol{x}_\theta$ by

$$
\boldsymbol{s}_\theta(\boldsymbol{x}_t, t, \boldsymbol{x}_T) = -\frac{\boldsymbol{x}_t - a_t \boldsymbol{x}_T - b_t \boldsymbol{x}_\theta(\boldsymbol{x}_t, t, \boldsymbol{x}_T)}{c_t^2}
\tag{31}
$$

Define $d_0 = 1$, the loss $\mathcal{J}^{(\rho)}(\theta)$ is further simplified to

$$
\begin{aligned}
&\mathcal{J}^{(\rho)}(\theta) - C \\
&= \sum_{n=0}^{N-1} \mathbb{E}_{q(\boldsymbol{x}_T) q^{(\rho)}(\boldsymbol{x}_0, \boldsymbol{x}_{t_{n+1}}|\boldsymbol{x}_T)} \left[ \frac{d_n^2 \|\boldsymbol{x}_\theta(\boldsymbol{x}_{t_{n+1}}, t_{n+1}, \boldsymbol{x}_T) - \boldsymbol{x}_0\|_2^2}{2\rho_n^2} \right] \\
&= \sum_{n=1}^{N} \frac{d_{n-1}^2}{2\rho_{n-1}^2} \mathbb{E}_{q(\boldsymbol{x}_T) q(\boldsymbol{x}_0|\boldsymbol{x}_T) q(\boldsymbol{x}_{t_n}|\boldsymbol{x}_0, \boldsymbol{x}_T)} \left[ \|\boldsymbol{x}_\theta(\boldsymbol{x}_{t_n}, t_n, \boldsymbol{x}_T) - \boldsymbol{x}_0\|_2^2 \right] \\
&= \sum_{n=1}^{N} \frac{d_{n-1}^2 c_{t_n}^4}{2\rho_{n-1}^2 b_{t_n}^2} \mathbb{E}_{q(\boldsymbol{x}_T) q(\boldsymbol{x}_0|\boldsymbol{x}_T) q(\boldsymbol{x}_{t_n}|\boldsymbol{x}_0, \boldsymbol{x}_T)} \left[ \|\boldsymbol{s}_\theta(\boldsymbol{x}_{t_n}, t_n, \boldsymbol{x}_T) - \nabla_{\boldsymbol{x}_{t_n}} \log q(\boldsymbol{x}_{t_n}|\boldsymbol{x}_0, \boldsymbol{x}_T)\|_2^2 \right]
\end{aligned}
\tag{32}
$$

Compared to the training objective of DDBMs in Eqn. (9), $\mathcal{J}^{(\rho)}(\theta)$ is totally equivalent up to a constant, by concentrating on the discretized timesteps $\{t_n\}_{n=1}^N$, choosing $q(\boldsymbol{x}_T)q(\boldsymbol{x}_0|\boldsymbol{x}_T)$ as the paired data distribution and using the weighting function $\gamma$ that satisfies $\gamma(t_n) = \frac{d_{n-1}^2 c_{t_n}^4}{2\rho_{n-1}^2 b_{t_n}^2}$. $\square$

### B.3 PROOF OF PROPOSITION 4.1

*Proof.* We first represent the PF-ODE (Eqn. (8))

$$
\mathrm{d}\boldsymbol{x}_t = \left[ f(t)\boldsymbol{x}_t - g^2(t) \left( \frac{1}{2} \nabla_{\boldsymbol{x}_t} \log q(\boldsymbol{x}_t|\boldsymbol{x}_T) - \nabla_{\boldsymbol{x}_t} \log q_{T|t}(\boldsymbol{x}_T|\boldsymbol{x}_t) \right) \right] \mathrm{d}t
\tag{33}
$$

with the data predictor $\boldsymbol{x}_\theta(\boldsymbol{x}_t, t, \boldsymbol{x}_T)$. We replace the bridge score $\nabla_{\boldsymbol{x}_t} \log q(\boldsymbol{x}_t|\boldsymbol{x}_T)$ with the network $\boldsymbol{s}_\theta(\boldsymbol{x}_t, t, \boldsymbol{x}_T)$, which is related to $\boldsymbol{x}_\theta(\boldsymbol{x}_t, t, \boldsymbol{x}_T)$ by Eqn. (14). Besides, $\nabla_{\boldsymbol{x}_t} \log q_{T|t}(\boldsymbol{x}_T|\boldsymbol{x}_t)$

can be analytically computed as

$$
\begin{aligned}
\nabla_{\boldsymbol{x}_t} \log q_{T|t}(\boldsymbol{x}_T|\boldsymbol{x}_t) &= \nabla_{\boldsymbol{x}_t} \log \frac{q(\boldsymbol{x}_t|\boldsymbol{x}_0, \boldsymbol{x}_T)q(\boldsymbol{x}_T|\boldsymbol{x}_0)}{q(\boldsymbol{x}_t|\boldsymbol{x}_0)} \\
&= \nabla_{\boldsymbol{x}_t} \log q(\boldsymbol{x}_t|\boldsymbol{x}_0, \boldsymbol{x}_T) - \nabla_{\boldsymbol{x}_t} \log q(\boldsymbol{x}_t|\boldsymbol{x}_0) \\
&= -\frac{\boldsymbol{x}_t - a_t\boldsymbol{x}_T - b_t\boldsymbol{x}_0}{c_t^2} + \frac{\boldsymbol{x}_t - \alpha_t\boldsymbol{x}_0}{\sigma_t^2} \\
&= -\frac{\frac{\mathrm{SNR}_T}{\mathrm{SNR}_t}(\boldsymbol{x}_t - \frac{\alpha_t}{\alpha_T}\boldsymbol{x}_T)}{\sigma_t^2(1 - \frac{\mathrm{SNR}_T}{\mathrm{SNR}_t})} \\
&= -\frac{a_t(\frac{\alpha_T}{\alpha_t}\boldsymbol{x}_t - \boldsymbol{x}_T)}{c_t^2}
\end{aligned}
\tag{34}
$$

Substituting Eqn. (14) and Eqn. (34) into Eqn. (33), the PF-ODE is transformed to

$$
\begin{aligned}
\mathrm{d}\boldsymbol{x}_t &= \left[ f(t)\boldsymbol{x}_t - g^2(t)\left( -\frac{\boldsymbol{x}_t - a_t\boldsymbol{x}_T - b_t\boldsymbol{x}_\theta(\boldsymbol{x}_t, t, \boldsymbol{x}_T)}{2c_t^2} + \frac{a_t(\frac{\alpha_T}{\alpha_t}\boldsymbol{x}_t - \boldsymbol{x}_T)}{c_t^2} \right) \right] \mathrm{d}t \\
&= \left[ \left( f(t) + g^2(t)\frac{1 - 2a_t\frac{\alpha_T}{\alpha_t}}{2c_t^2} \right) \boldsymbol{x}_t + g^2(t)\frac{a_t}{2c_t^2}\boldsymbol{x}_T - g^2(t)\frac{b_t}{2c_t^2}\boldsymbol{x}_\theta(\boldsymbol{x}_t, t, \boldsymbol{x}_T) \right] \mathrm{d}t \\
&= \left[ \left( f(t) + \frac{g^2(t)}{\sigma_t^2} - \frac{g^2(t)}{2c_t^2} \right) \boldsymbol{x}_t + g^2(t)\frac{a_t}{2c_t^2}\boldsymbol{x}_T - g^2(t)\frac{b_t}{2c_t^2}\boldsymbol{x}_\theta(\boldsymbol{x}_t, t, \boldsymbol{x}_T) \right] \mathrm{d}t
\end{aligned}
\tag{35}
$$

On the other hand, the ODE corresponding to DBIMs (Eqn. (16)) can be expanded as

$$
\frac{\mathrm{d}\boldsymbol{x}_t}{c_t} - \frac{c_t'}{c_t^2}\boldsymbol{x}_t\mathrm{d}t = \left[ \left( \frac{a_t}{c_t} \right)' \boldsymbol{x}_T + \left( \frac{b_t}{c_t} \right)' \boldsymbol{x}_\theta(\boldsymbol{x}_t, t, \boldsymbol{x}_T) \right] \mathrm{d}t
\tag{36}
$$

where we have denoted $(\cdot)' := \frac{\mathrm{d}(\cdot)}{\mathrm{d}t}$. Further simplification gives

$$
\mathrm{d}\boldsymbol{x}_t = \left[ \frac{c_t'}{c_t}\boldsymbol{x}_t + \left( a_t' - a_t\frac{c_t'}{c_t} \right) \boldsymbol{x}_T + \left( b_t' - b_t\frac{c_t'}{c_t} \right) \boldsymbol{x}_\theta(\boldsymbol{x}_t, t, \boldsymbol{x}_T) \right] \mathrm{d}t
\tag{37}
$$

The coefficients $a_t, b_t, c_t$ are determined by the noise schedule $\alpha_t, \sigma_t$ in diffusion models. Computing their derivatives will produce terms involving $f(t), g(t)$, which are used to define the forward SDE. As revealed in diffusion models, $f(t), g(t)$ are related to $\alpha_t, \sigma_t$ by $f(t) = \frac{\mathrm{d}\log\alpha_t}{\mathrm{d}t}$, $g^2(t) = \frac{\mathrm{d}\sigma_t^2}{\mathrm{d}t} - 2\frac{\mathrm{d}\log\alpha_t}{\mathrm{d}t}\sigma_t^2$. We can derive the reverse relation of $\alpha_t, \sigma_t$ and $f(t), g(t)$:

$$
\alpha_t = e^{\int_0^t f(\tau)\mathrm{d}\tau}, \quad \sigma_t^2 = \alpha_t^2 \int_0^t \frac{g^2(\tau)}{\alpha_\tau^2}\mathrm{d}\tau
\tag{38}
$$

which can facilitate subsequent calculation. We first compute the derivative of a common term in $a_t, b_t, c_t$:

$$
\left( \frac{1}{\mathrm{SNR}_t} \right)' = \left( \frac{\sigma_t^2}{\alpha_t^2} \right)' = \frac{g^2(t)}{\alpha_t^2}
\tag{39}
$$

For $c_t$, since $c_t^2 = \sigma_t^2(1 - \frac{\mathrm{SNR}_T}{\mathrm{SNR}_t})$, we have

$$
\frac{c_t'}{c_t} = (\log c_t)' = \frac{1}{2}(\log c_t^2)' = \frac{1}{2}(\log\sigma_t^2 + \log(1 - \frac{\mathrm{SNR}_T}{\mathrm{SNR}_t}))'
\tag{40}
$$

where

$$
(\log\sigma_t^2)' = (\log\frac{\sigma_t^2}{\alpha_t^2})' + (\log\alpha_t^2)' = \frac{g^2(t)}{\alpha_t^2}\frac{\alpha_t^2}{\sigma_t^2} + 2f(t) = \frac{g^2(t)}{\sigma_t^2} + 2f(t)
\tag{41}
$$

and

$$
(\log(1 - \frac{\mathrm{SNR}_T}{\mathrm{SNR}_t}))' = -\frac{\mathrm{SNR}_T}{1 - \frac{\mathrm{SNR}_T}{\mathrm{SNR}_t}}\left( \frac{1}{\mathrm{SNR}_t} \right)' = -\frac{\mathrm{SNR}_T}{c_t^2}\sigma_t^2\frac{g^2(t)}{\alpha_t^2} = -\frac{g^2(t)}{c_t^2}\frac{\mathrm{SNR}_T}{\mathrm{SNR}_t}
\tag{42}
$$

Substituting Eqn. (41) and Eqn. (42) into Eqn. (40), and using the relation $\frac{\text{SNR}_T}{\text{SNR}_t} = 1 - \frac{c_t^2}{\sigma_t^2}$, we have

$$\frac{c_t'}{c_t} = f(t) + \frac{g^2(t)}{2\sigma_t^2} - \frac{g^2(t)}{2c_t^2}\frac{\text{SNR}_T}{\text{SNR}_t} = f(t) + \frac{g^2(t)}{\sigma_t^2} - \frac{g^2(t)}{2c_t^2} \tag{43}$$

For $a_t$, since $a_t = \frac{\alpha_t}{\alpha_T}\frac{\text{SNR}_T}{\text{SNR}_t}$, we have

$$\frac{a_t'}{a_t} = (\log a_t)' = (\log \alpha_t)' + (\log\frac{\text{SNR}_T}{\text{SNR}_t})' = f(t) + \text{SNR}_t\frac{g^2(t)}{\alpha_t^2} = f(t) + \frac{g^2(t)}{\sigma_t^2} \tag{44}$$

For $b_t$, since $b_t = \alpha_t(1 - \frac{\text{SNR}_T}{\text{SNR}_t})$, we have

$$\frac{b_t'}{b_t} = (\log b_t)' = (\log \alpha_t)' + (\log(1-\frac{\text{SNR}_T}{\text{SNR}_t}))' = f(t) - \frac{g^2(t)}{c_t^2}\frac{\text{SNR}_T}{\text{SNR}_t} = f(t) + \frac{g^2(t)}{\sigma_t^2} - \frac{g^2(t)}{c_t^2} \tag{45}$$

Therefore,

$$a_t' - a_t\frac{c_t'}{c_t} = a_t(\frac{a_t'}{a_t} - \frac{c_t'}{c_t}) = \frac{g^2(t)}{2c_t^2}a_t \tag{46}$$

and

$$b_t' - b_t\frac{c_t'}{c_t} = b_t(\frac{b_t'}{b_t} - \frac{c_t'}{c_t}) = -\frac{g^2(t)}{2c_t^2}b_t \tag{47}$$

Substituting Eqn. (43), Eqn. (46) and Eqn. (47) into the ODE of DBIMs in Eqn. (37), we obtain exactly the PF-ODE in Eqn. (35). □

## C   MORE THEORETICAL DISCUSSIONS

### C.1   MARKOV PROPERTY OF THE GENERALIZED DIFFUSION BRIDGES

We aim to analyze the Markov property of the forward process corresponding to our generalized diffusion bridge in Section 3.1. The forward process of $q^{(\rho)}$ can be induced by Bayes' rule as

$$q^{(\rho)}(\boldsymbol{x}_{t_{n+1}}|\boldsymbol{x}_0, \boldsymbol{x}_{t_n}, \boldsymbol{x}_T) = \frac{q^{(\rho)}(\boldsymbol{x}_{t_n}|\boldsymbol{x}_0, \boldsymbol{x}_{t_{n+1}}, \boldsymbol{x}_T)q^{(\rho)}(\boldsymbol{x}_{t_{n+1}}|\boldsymbol{x}_0, \boldsymbol{x}_T)}{q^{(\rho)}(\boldsymbol{x}_{t_n}|\boldsymbol{x}_0, \boldsymbol{x}_T)} \tag{48}$$

where $q^{(\rho)}(\boldsymbol{x}_t|\boldsymbol{x}_0, \boldsymbol{x}_T) = q(\boldsymbol{x}_t|\boldsymbol{x}_0, \boldsymbol{x}_T)$ is the marginal distribution of the diffusion bridge in Eqn. (6), and $q^{(\rho)}(\boldsymbol{x}_{t_n}|\boldsymbol{x}_0, \boldsymbol{x}_{t_{n+1}}, \boldsymbol{x}_T)$ is defined in Eqn. (11) as

$$q^{(\rho)}(\boldsymbol{x}_{t_n}|\boldsymbol{x}_0, \boldsymbol{x}_{t_{n+1}}, \boldsymbol{x}_T) = \mathcal{N}(a_{t_n}\boldsymbol{x}_T + b_{t_n}\boldsymbol{x}_0 + \sqrt{c_{t_n}^2 - \rho_n^2}\frac{\boldsymbol{x}_{t_{n+1}} - a_{t_{n+1}}\boldsymbol{x}_T - b_{t_{n+1}}\boldsymbol{x}_0}{c_{t_{n+1}}}, \rho_n^2\boldsymbol{I}). \tag{49}$$

Due to the marginal preservation property (Proposition 3.1), we have $q^{(\rho)}(\boldsymbol{x}_{t_{n+1}}|\boldsymbol{x}_0, \boldsymbol{x}_T) = q(\boldsymbol{x}_{t_{n+1}}|\boldsymbol{x}_0, \boldsymbol{x}_T)$ and $q^{(\rho)}(\boldsymbol{x}_{t_n}|\boldsymbol{x}_0, \boldsymbol{x}_T) = q(\boldsymbol{x}_{t_n}|\boldsymbol{x}_0, \boldsymbol{x}_T)$, where $q(\boldsymbol{x}_t|\boldsymbol{x}_0, \boldsymbol{x}_T) = \mathcal{N}(a_t\boldsymbol{x}_T + b_t\boldsymbol{x}_0, c_t^2\boldsymbol{I})$ is the forward transition kernel in Eqn. (6). To identify whether $q^{(\rho)}(\boldsymbol{x}_{t_{n+1}}|\boldsymbol{x}_0, \boldsymbol{x}_{t_n}, \boldsymbol{x}_T)$ is Markovian, we only need to examine the dependence of $\boldsymbol{x}_{t_{n+1}}$ on $\boldsymbol{x}_0$. To this end, we proceed to derive conditions under which $\nabla_{\boldsymbol{x}_{t_{n+1}}} \log q^{(\rho)}(\boldsymbol{x}_{t_{n+1}}|\boldsymbol{x}_0, \boldsymbol{x}_{t_n}, \boldsymbol{x}_T)$ involves terms concerning $\boldsymbol{x}_0$.

Specifically, $\nabla_{\boldsymbol{x}_{t_{n+1}}} \log q^{(\rho)}(\boldsymbol{x}_{t_{n+1}}|\boldsymbol{x}_0, \boldsymbol{x}_{t_n}, \boldsymbol{x}_T)$ can be calculated as

$$\begin{aligned}
&\nabla_{\boldsymbol{x}_{t_{n+1}}} \log q^{(\rho)}(\boldsymbol{x}_{t_{n+1}}|\boldsymbol{x}_0, \boldsymbol{x}_{t_n}, \boldsymbol{x}_T)\\
=&\nabla_{\boldsymbol{x}_{t_{n+1}}} \log q^{(\rho)}(\boldsymbol{x}_{t_n}|\boldsymbol{x}_0, \boldsymbol{x}_{t_{n+1}}, \boldsymbol{x}_T) + \nabla_{\boldsymbol{x}_{t_{n+1}}} \log q^{(\rho)}(\boldsymbol{x}_{t_{n+1}}|\boldsymbol{x}_0, \boldsymbol{x}_T)\\
=&-\frac{\sqrt{c_{t_n}^2 - \rho_n^2}(a_{t_n}\boldsymbol{x}_T + b_{t_n}\boldsymbol{x}_0 + \sqrt{c_{t_n}^2 - \rho_n^2}\frac{\boldsymbol{x}_{t_{n+1}} - a_{t_{n+1}}\boldsymbol{x}_T - b_{t_{n+1}}\boldsymbol{x}_0}{c_{t_{n+1}}} - \boldsymbol{x}_{t_n})}{c_{t_{n+1}}\rho_n^2}\\
&-\frac{\boldsymbol{x}_{t_{n+1}} - a_{t_{n+1}}\boldsymbol{x}_T - b_{t_{n+1}}\boldsymbol{x}_0}{c_{t_{n+1}}^2}\\
=&\frac{b_{t_{n+1}}c_{t_n}^2 - b_{t_n}c_{t_{n+1}}\sqrt{c_{t_n}^2 - \rho_n^2}}{c_{t_{n+1}}^2\rho_n^2}\boldsymbol{x}_0 + C(\boldsymbol{x}_{t_n}, \boldsymbol{x}_{t_{n+1}}, \boldsymbol{x}_T)
\end{aligned} \tag{50}$$

where $C(\boldsymbol{x}_{t_n}, \boldsymbol{x}_{t_{n+1}}, \boldsymbol{x}_T)$ are terms irrelevant to $\boldsymbol{x}_0$. Therefore,

$$q^{(\rho)}(\boldsymbol{x}_{t_{n+1}}|\boldsymbol{x}_0, \boldsymbol{x}_{t_n}, \boldsymbol{x}_T) \text{ is Markovian} \iff \frac{b_{t_{n+1}} c_{t_n}^2 - b_{t_n} c_{t_{n+1}} \sqrt{c_{t_n}^2 - \rho_n^2}}{c_{t_{n+1}}^2 \rho_n^2} = 0$$

$$\iff \rho_n = \sigma_{t_n} \sqrt{1 - \frac{\text{SNR}_{t_{n+1}}}{\text{SNR}_{t_n}}}$$

(51)

which is exactly a boundary choice of the variance parameter $\rho$. Under the other boundary choice $\rho_n = 0$ and intermediate ones satisfying $0 < \rho_n < \sigma_{t_n} \sqrt{1 - \frac{\text{SNR}_{t_{n+1}}}{\text{SNR}_{t_n}}}$, the forward process $q^{(\rho)}(\boldsymbol{x}_{t_{n+1}}|\boldsymbol{x}_0, \boldsymbol{x}_{t_n}, \boldsymbol{x}_T)$ is non-Markovian.

## C.2 THE INSIGNIFICANCE OF LOSS WEIGHTING IN TRAINING

The insignificance of the weighting mismatch in Proposition 3.2 can be interpreted from two aspects. On the one hand, $\mathcal{L}$ consists of independent terms concerning individual timesteps (as long as the network's parameters are not shared across different $t$), ensuring that the global minimum remains the same as minimizing the loss at each timestep, regardless of the weighting. On the other hand, $\mathcal{L}$ under different weightings are mutually bounded by $\frac{\min_t w_t}{\max_t \gamma_t} \mathcal{L}_\gamma(\theta) \leq \mathcal{L}_w(\theta) \leq \frac{\max_t w_t}{\min_t \gamma_t} \mathcal{L}_\gamma(\theta)$. Besides, it is widely acknowledged that in diffusion models, the weighting corresponding to variational inference may yield superior likelihood but suboptimal sample quality (Ho et al., 2020; Song et al., 2021c), which is not preferred in practice.

## C.3 SPECIAL CASES AND RELATIONSHIP WITH PRIOR WORKS

**Connection to Flow Matching** When the noise schedule $\alpha_t = 1, T = 1$ and $\sigma_t = \sqrt{\beta t}$, the forward process becomes $q(\boldsymbol{x}_t|\boldsymbol{x}_0, \boldsymbol{x}_1) = \mathcal{N}(t\boldsymbol{x}_T + (1-t)\boldsymbol{x}_0, \beta t(1-t))$ which is a Brownian bridge. When $\beta \to 0$, there will be no intermediate noise and the forward process is similar to flow matching (Lipman et al., 2022; Albergo et al., 2023). In this limit, the DBIM ($\eta = 1$) updating rule from time $t$ to time $s < t$ will become $\boldsymbol{x}_s = s\boldsymbol{x}_T + (1-s)\boldsymbol{x}_\theta(\boldsymbol{x}_t, t, \boldsymbol{x}_T) = \frac{s}{t}\boldsymbol{x}_t + (1 - \frac{s}{t})\boldsymbol{x}_\theta(\boldsymbol{x}_t, t) = \boldsymbol{x}_t - (t-s)\boldsymbol{v}_\theta(\boldsymbol{x}_t, t)$. Here we define $\boldsymbol{v}_\theta(\boldsymbol{x}_t, t) := \frac{\boldsymbol{x}_t - \boldsymbol{x}_\theta(\boldsymbol{x}_t, t)}{t}$ as the velocity function of the probability flow (i.e., the drift of the ODE) in flow matching methods. Therefore, in the flow matching case, DBIM is a simple Euler step of the flow.

**Connection to DDIM** In the regions where $t$ is small and $\frac{\text{SNR}_T}{\text{SNR}_t}$ is close to 0, we have $a_t \approx 0, b_t \approx \alpha_t, c_t \approx \sigma_t$. Therefore, the forward process of DDBM in this case is approximately $q(\boldsymbol{x}_t|\boldsymbol{x}_0, \boldsymbol{x}_T) = \mathcal{N}(\alpha_t \boldsymbol{x}_0, \sigma_t^2 \boldsymbol{I})$, which is the forward process of the corresponding diffusion model. Moreover, in this case, the DBIM ($\eta = 0$) step is approximately

$$\boldsymbol{x}_s \approx \frac{\sigma_s}{\sigma_t} \boldsymbol{x}_t + \sigma_s \left( \frac{\alpha_s}{\sigma_s} - \frac{\alpha_t}{\sigma_t} \right) \boldsymbol{x}_\theta(\boldsymbol{x}_t, t, \boldsymbol{x}_T)$$

(52)

which is exactly DDIM (Song et al., 2021a), except that the data prediction network $\boldsymbol{x}_\theta$ is dependent on $\boldsymbol{x}_T$. This indicates that when $t$ is small so that the component of $x_T$ in $x_t$ is negligible, DBIM recovers DDIM.

**Connection to Posterior Sampling** The previous work I$^2$SB (Liu et al., 2023b) also employs diffusion bridges with discrete timesteps, though their noise schedule is restricted to the variance exploding (VE) type with $f(t) = 0$ in the forward process. For generation, they adopt a similar approach to DDPM (Ho et al., 2020) by iterative sampling from the posterior distribution $p_\theta(\boldsymbol{x}_{n-1}|\boldsymbol{x}_n)$, which is a parameterized and shortened diffusion bridge between the endpoints $\hat{\boldsymbol{x}}_0 = \boldsymbol{x}_\theta(\boldsymbol{x}_n, t_n, \boldsymbol{x}_N)$ and $\boldsymbol{x}_n$. Since the posterior distribution is not conditioned on $\boldsymbol{x}_T$ (except through the parameterized network), the corresponding forward diffusion bridge is Markovian. Thus, the posterior sampling in I$^2$SB is a special case of DBIM by setting $\eta = 0$ and $f(t) = 0$.

## C.4 PERSPECTIVE OF EXPONENTIAL INTEGRATORS

Exponential integrators (Calvo & Palencia, 2006; Hochbruck et al., 2009) are widely adopted in recent works concerning fast sampling of diffusion models (Zhang & Chen, 2022; Zheng et al.,

2023a; Gonzalez et al., 2023). Suppose we have an ODE

$$\mathrm{d}\boldsymbol{x}_t = [a(t)\boldsymbol{x}_t + b(t)\boldsymbol{F}_\theta(\boldsymbol{x}_t, t)]\mathrm{d}t \tag{53}$$

where $\boldsymbol{F}_\theta$ is the parameterized prediction function that we want to approximate with Taylor expansion. The usual way of representing its analytic solution $\boldsymbol{x}_t$ at time $t$ with respect to an initial condition $\boldsymbol{x}_s$ at time $s$ is

$$\boldsymbol{x}_t = \boldsymbol{x}_s + \int_s^t [a(\tau)\boldsymbol{x}_\tau + b(\tau)\boldsymbol{F}_\theta(\boldsymbol{x}_\tau, \tau)]\mathrm{d}\tau \tag{54}$$

By approximating the involved integrals in Eqn. (54), we can obtain direct discretizations of Eqn. (53) such as Euler's method. The key insight of exponential integrators is that, it is often better to utilize the "semi-linear" structure of Eqn. (53) and analytically cancel the linear term $a(t)\boldsymbol{x}_t$. This way, we can obtain solutions that only involve integrals of $\boldsymbol{F}_\theta$ and result in lower discretization errors. Specifically, by the "variation-of-constants" formula, the exact solution of Eqn. (53) can be alternatively given by

$$\boldsymbol{x}_t = e^{\int_s^t a(\tau)\mathrm{d}\tau}\boldsymbol{x}_s + \int_s^t e^{\int_\tau^t a(r)\mathrm{d}r}b(\tau)\boldsymbol{F}_\theta(\boldsymbol{x}_\tau, \tau)\mathrm{d}\tau \tag{55}$$

We can apply this transformation to the PF-ODE in DDBMs. By collecting the linear terms w.r.t. $\boldsymbol{x}_t$, Eqn. (8) can be rewritten as (already derived in Appendix B.3)

$$\mathrm{d}\boldsymbol{x}_t = \left[\left(f(t) + \frac{g^2(t)}{\sigma_t^2} - \frac{g^2(t)}{2c_t^2}\right)\boldsymbol{x}_t + g^2(t)\frac{a_t}{2c_t^2}\boldsymbol{x}_T - g^2(t)\frac{b_t}{2c_t^2}\boldsymbol{x}_\theta(\boldsymbol{x}_t, t, \boldsymbol{x}_T)\right]\mathrm{d}t \tag{56}$$

By corresponding it to Eqn. (53), we have

$$a(t) = f(t) + \frac{g^2(t)}{\sigma_t^2} - \frac{g^2(t)}{2c_t^2}, \quad b_1(t) = g^2(t)\frac{a_t}{2c_t^2}, \quad b_2(t) = -g^2(t)\frac{b_t}{2c_t^2} \tag{57}$$

From Eqn. (43), Eqn. (46) and Eqn. (47), we know

$$a(t) = \frac{\mathrm{d}\log c_t}{\mathrm{d}t}, \quad b_1(t) = a_t\frac{\mathrm{d}\log(a_t/c_t)}{\mathrm{d}t}, \quad b_2(t) = b_t\frac{\mathrm{d}\log(b_t/c_t)}{\mathrm{d}t} \tag{58}$$

Note that these relations are known in advance when converting from our ODE to the PF-ODE. Otherwise, finding them in this inverse conversion will be challenging. The integrals in Eqn. (55) can then be calculated as $\int_s^t a(\tau)\mathrm{d}\tau = \log c_t - \log c_s$. Thus

$$e^{\int_s^t a(\tau)\mathrm{d}\tau} = \frac{c_t}{c_s}, \quad e^{\int_\tau^t a(r)\mathrm{d}r} = \frac{c_t}{c_\tau} \tag{59}$$

Therefore, the exact solution in Eqn. (55) becomes

$$\begin{aligned}\boldsymbol{x}_t &= \frac{c_t}{c_s}\boldsymbol{x}_s + c_t\int_s^t \frac{a_\tau}{c_\tau}\boldsymbol{x}_T\mathrm{d}\log\left(\frac{a_\tau}{c_\tau}\right) + c_t\int_s^t \frac{b_\tau}{c_\tau}\boldsymbol{x}_\theta(\boldsymbol{x}_\tau, \tau, \boldsymbol{x}_T)\mathrm{d}\log\left(\frac{b_\tau}{c_\tau}\right) \\ &= \frac{c_t}{c_s}\boldsymbol{x}_s + \left(a_t - \frac{c_t}{c_s}a_s\right)\boldsymbol{x}_T + c_t\int_s^t \frac{b_\tau}{c_\tau}\boldsymbol{x}_\theta(\boldsymbol{x}_\tau, \tau, \boldsymbol{x}_T)\mathrm{d}\log\left(\frac{b_\tau}{c_\tau}\right)\end{aligned} \tag{60}$$

which is the same as Eqn. (17) after exchanging $s$ and $t$ and changing the time variable in the integral to $\lambda_t = \log\left(\frac{b_t}{c_t}\right)$.

Lastly, we emphasize the advantage of DBIM over employing exponential integrators. First, deriving our ODE via exponential integrators requires the PF-ODE as preliminary. However, the PF-ODE alone cannot handle the singularity at the start point and presents theoretical challenges. Moreover, the conversion process from the PF-ODE to our ODE is intricate, while DBIM retains the overall simplicity. Additionally, DBIM supports varying levels of stochasticity during sampling, unlike the deterministic nature of ODE-based methods. This stochasticity can mitigate sampling errors via the Langevin mechanism (Song et al., 2021c), potentially enhancing the generation quality.

## D   DERIVATION OF OUR HIGH-ORDER NUMERICAL SOLVERS

High-order solvers of Eqn. (17) can be developed by approximating $\boldsymbol{x}_\theta$ in the integral with Taylor expansion. Specifically, as a function of $\lambda$, we have $\boldsymbol{x}_\theta(\boldsymbol{x}_{t_\lambda}, t_\lambda, \boldsymbol{x}_T) \approx \boldsymbol{x}_\theta(\boldsymbol{x}_t, t, \boldsymbol{x}_T) + \sum_{k=1}^n \frac{(\lambda - \lambda_t)^k}{k!} \boldsymbol{x}_\theta^{(k)}(\boldsymbol{x}_t, t, \boldsymbol{x}_T)$, where $\boldsymbol{x}_\theta^{(k)}(\boldsymbol{x}_t, t, \boldsymbol{x}_T) = \frac{\mathrm{d}^k \boldsymbol{x}_\theta(\boldsymbol{x}_{t_\lambda}, t_\lambda, \boldsymbol{x}_T)}{\mathrm{d}\lambda^k}\Big|_{\lambda = \lambda_t}$ is the $k$-th order derivative w.r.t. $\lambda$, which can be estimated with finite difference of past network outputs.

**2nd-Order Case**   With the Taylor expansion $\boldsymbol{x}_\theta(\boldsymbol{x}_{t_\lambda}, t_\lambda, \boldsymbol{x}_T) \approx \boldsymbol{x}_\theta(\boldsymbol{x}_t, t, \boldsymbol{x}_T) + (\lambda - \lambda_t)\boldsymbol{x}_\theta^{(1)}(\boldsymbol{x}_t, t, \boldsymbol{x}_T)$, we have

$$
\begin{aligned}
\int_{\lambda_t}^{\lambda_s} e^\lambda \boldsymbol{x}_\theta(\boldsymbol{x}_{t_\lambda}, t_\lambda, \boldsymbol{x}_T)\mathrm{d}\lambda &\approx \left(\int_{\lambda_t}^{\lambda_s} e^\lambda \mathrm{d}\lambda\right) \boldsymbol{x}_\theta(\boldsymbol{x}_t, t, \boldsymbol{x}_T) + \left(\int_{\lambda_t}^{\lambda_s} (\lambda - \lambda_t)e^\lambda \mathrm{d}\lambda\right) \boldsymbol{x}_\theta^{(1)}(\boldsymbol{x}_t, t, \boldsymbol{x}_T) \\
&\approx e^{\lambda_s}\left[(1 - e^{-h})\hat{\boldsymbol{x}}_t + (h - 1 + e^{-h})\hat{\boldsymbol{x}}_t^{(1)}\right]
\end{aligned}
\tag{61}
$$

where we use $\hat{\boldsymbol{x}}_t$ to denote the network output at time $t$, and $h = \lambda_s - \lambda_t > 0$. Suppose we have used a previous timestep $u$ ($s < t < u$), the first-order derivative can be estimated by

$$
\hat{\boldsymbol{x}}_t^{(1)} \approx \frac{\hat{\boldsymbol{x}}_t - \hat{\boldsymbol{x}}_u}{h_1}, \quad h_1 = \lambda_t - \lambda_u
\tag{62}
$$

**3rd-Order Case**   With the Taylor expansion $\boldsymbol{x}_\theta(\boldsymbol{x}_{t_\lambda}, t_\lambda, \boldsymbol{x}_T) \approx \boldsymbol{x}_\theta(\boldsymbol{x}_t, t, \boldsymbol{x}_T) + (\lambda - \lambda_t)\boldsymbol{x}_\theta^{(1)}(\boldsymbol{x}_t, t, \boldsymbol{x}_T) + \frac{(\lambda - \lambda_t)^2}{2}\boldsymbol{x}_\theta^{(2)}(\boldsymbol{x}_t, t, \boldsymbol{x}_T)$, we have

$$
\begin{aligned}
&\int_{\lambda_t}^{\lambda_s} e^\lambda \boldsymbol{x}_\theta(\boldsymbol{x}_{t_\lambda}, t_\lambda, \boldsymbol{x}_T)\mathrm{d}\lambda \\
&\approx \left(\int_{\lambda_t}^{\lambda_s} e^\lambda \mathrm{d}\lambda\right) \boldsymbol{x}_\theta(\boldsymbol{x}_t, t, \boldsymbol{x}_T) + \left(\int_{\lambda_t}^{\lambda_s} (\lambda - \lambda_t)e^\lambda \mathrm{d}\lambda\right) \boldsymbol{x}_\theta^{(1)}(\boldsymbol{x}_t, t, \boldsymbol{x}_T) \\
&\quad + \left(\int_{\lambda_t}^{\lambda_s} \frac{(\lambda - \lambda_t)^2}{2} e^\lambda \mathrm{d}\lambda\right) \boldsymbol{x}_\theta^{(2)}(\boldsymbol{x}_t, t, \boldsymbol{x}_T) \\
&\approx e^{\lambda_s}\left[(1 - e^{-h})\hat{\boldsymbol{x}}_t + (h - 1 + e^{-h})\hat{\boldsymbol{x}}_t^{(1)} + \left(\frac{h^2}{2} - h + 1 - e^{-h}\right)\hat{\boldsymbol{x}}_t^{(2)}\right]
\end{aligned}
\tag{63}
$$

Similarly, suppose we have two previous timesteps $u_1, u_2$ ($s < t < u_1 < u_2$), and denote $h_1 := \lambda_t - \lambda_{u_1}, h_2 := \lambda_{u_1} - \lambda_{u_2}$, the first-order and second-order derivatives can be estimated by

$$
\hat{\boldsymbol{x}}_t^{(1)} \approx \frac{\frac{\hat{\boldsymbol{x}}_t - \hat{\boldsymbol{x}}_{u_1}}{h_1}(2h_1 + h_2) - \frac{\hat{\boldsymbol{x}}_{u_1} - \hat{\boldsymbol{x}}_{u_2}}{h_2} h_1}{h_1 + h_2}, \quad \hat{\boldsymbol{x}}_t^{(2)} \approx 2\frac{\frac{\hat{\boldsymbol{x}}_t - \hat{\boldsymbol{x}}_{u_1}}{h_1} - \frac{\hat{\boldsymbol{x}}_{u_1} - \hat{\boldsymbol{x}}_{u_2}}{h_2}}{h_1 + h_2}
\tag{64}
$$

The high-order samplers for DDBMs also theoretically guarantee the order of convergence, similar to those for diffusion models (Zhang & Chen, 2022; Lu et al., 2022b; Zheng et al., 2023a). We omit the proofs here as they deviate from our main contributions.

# E ALGORITHM

---

**Algorithm 1** DBIM (high-order)

---

**Require:** condition $\boldsymbol{x}_T$, timesteps $0 \leq t_0 < t_1 < \cdots < t_{N-1} < t_N = T$, data prediction model $\boldsymbol{x}_\theta$, booting noise $\boldsymbol{\epsilon} \sim \mathcal{N}(\boldsymbol{0}, \boldsymbol{I})$, noise schedule $a_t, b_t, c_t, \lambda_t = \log(b_t/c_t)$, order $o$ (2 or 3).
1: $\hat{\boldsymbol{x}}_T \leftarrow \boldsymbol{x}_\theta(\boldsymbol{x}_T, T, \boldsymbol{x}_T)$
2: $\boldsymbol{x}_{t_{N-1}} \leftarrow a_t \boldsymbol{x}_T + b_t \hat{\boldsymbol{x}}_T + c_t \boldsymbol{\epsilon}$
3: **for** $i \leftarrow N - 1$ to 1 **do**
4:     $s, t \leftarrow t_{i-1}, t_i; h \leftarrow \lambda_s - \lambda_t$
5:     $\hat{\boldsymbol{x}}_t \leftarrow \boldsymbol{x}_\theta(\boldsymbol{x}_t, t, \boldsymbol{x}_T)$
6:     **if** $o = 2$ or $i = N - 1$ **then**
7:         $u \leftarrow t_{i+1}; h_1 \leftarrow \lambda_t - \lambda_u$
8:         Estimate $\hat{\boldsymbol{x}}_t^{(1)}$ with Eqn. (62)
9:         $\hat{\boldsymbol{I}} \leftarrow e^{\lambda_s} \left[ (1 - e^{-h}) \hat{\boldsymbol{x}}_t + (h - 1 + e^{-h}) \hat{\boldsymbol{x}}_t^{(1)} \right]$
10:     **else**
11:         $u_1, u_2 \leftarrow t_{i+1}, t_{i+2}; h_1 \leftarrow \lambda_t - \lambda_{u_1}; h_2 \leftarrow \lambda_{u_1} - \lambda_{u_2}$
12:         Estimate $\hat{\boldsymbol{x}}_t^{(1)}, \hat{\boldsymbol{x}}_t^{(2)}$ with Eqn. (64)
13:         $\hat{\boldsymbol{I}} \leftarrow e^{\lambda_s} \left[ (1 - e^{-h}) \hat{\boldsymbol{x}}_t + (h - 1 + e^{-h}) \hat{\boldsymbol{x}}_t^{(1)} + \left( \frac{h^2}{2} - h + 1 - e^{-h} \right) \hat{\boldsymbol{x}}_t^{(2)} \right]$
14:     **end if**
15:     $\boldsymbol{x}_s \leftarrow \frac{c_s}{c_t} \boldsymbol{x}_t + \left( a_s - \frac{c_s}{c_t} a_t \right) \boldsymbol{x}_T + c_s \hat{\boldsymbol{I}}$
16: **end for**
17: **return** $\boldsymbol{x}_{t_0}$

---

# F EXPERIMENT DETAILS

## F.1 MODEL DETAILS

DDBMs and DBIMs are assessed using the same trained diffusion bridge models. For image translation tasks, we directly adopt the pretrained checkpoints provided by DDBMs. The data prediction model $\boldsymbol{x}_\theta(\boldsymbol{x}_t, t, \boldsymbol{x}_T)$ mentioned in the main text is parameterized by the network $\boldsymbol{F}_\theta$ as $\boldsymbol{x}_\theta(\boldsymbol{x}_t, t, \boldsymbol{x}_T) = c_{\text{skip}}(t)\boldsymbol{x}_t + c_{\text{out}}(t)\boldsymbol{F}_\theta(c_{\text{in}}(t)\boldsymbol{x}_t, c_{\text{noise}}(t), \boldsymbol{x}_T)$, where

$$c_{\text{in}}(t) = \frac{1}{\sqrt{a_t^2 \sigma_T^2 + b_t^2 \sigma_0^2 + 2a_t b_t \sigma_{0T} + c_t}}, \quad c_{\text{out}}(t) = \sqrt{a_t^2(\sigma_T^2 \sigma_0^2 - \sigma_{0T}^2) + \sigma_0^2 c_t} \, c_{\text{in}}(t)$$

$$c_{\text{skip}}(t) = (b_t \sigma_0^2 + a_t \sigma_{0T}) c_{\text{in}}^2(t), \quad c_{\text{noise}}(t) = \frac{1}{4} \log t \tag{65}$$

and

$$\sigma_0^2 = \text{Var}[\boldsymbol{x}_0], \sigma_T^2 = \text{Var}[\boldsymbol{x}_T], \sigma_{0T} = \text{Cov}[\boldsymbol{x}_0, \boldsymbol{x}_T] \tag{66}$$

For the image inpainting task on ImageNet 256×256 with 128×128 center mask, DDBMs do not provide available checkpoints. Therefore, we train a new model from scratch using the noise schedule of I²SB (Liu et al., 2023b). The network is initialized from the pretrained class-conditional diffusion model on ImageNet 256×256 provided by (Dhariwal & Nichol, 2021), while additionally conditioned on $\boldsymbol{x}_T$. The data prediction model in this case is parameterized by the network $\boldsymbol{F}_\theta$ as $\boldsymbol{x}_\theta(\boldsymbol{x}_t, t, \boldsymbol{x}_T) = \boldsymbol{x}_t - \sigma_t \boldsymbol{F}_\theta(\boldsymbol{x}_t, t, \boldsymbol{x}_T)$ and trained by minimizing the loss $\mathcal{L}(\theta) = \mathbb{E}_{t, \boldsymbol{x}_0, \boldsymbol{x}_T} \left[ \frac{1}{\sigma_t^2} \|\boldsymbol{x}_\theta(\boldsymbol{x}_t, t, \boldsymbol{x}_T) - \boldsymbol{x}_0\|_2^2 \right]$. We train the model on 8 NVIDIA A800 GPU cards with a batch size of 256 for 400k iterations, which takes around 19 days.

## F.2 SAMPLING DETAILS

We elaborate on the sampling configurations of different approaches, including the choice of timesteps $\{t_i\}_{i=0}^N$ and details of the samplers. In this work, we adopt $t_{\min} = 0.0001$ and $t_{\max} = 1$ following (Zhou et al., 2023). For the DDBM baseline, we use the hybrid, high-order Heun sampler

proposed in their work with an Euler step ratio of 0.33, which is the best performing configuration for the image-to-image translation task. We use the same timesteps distributed according to EDM (Karras et al., 2022)'s scheduling $(t_{\max}^{1/\rho} + \frac{i}{N}(t_{\min}^{1/\rho} - t_{\max}^{1/\rho}))^\rho$, consistent with the official implementation of DDBM. For DBIM, since the initial sampling step is distinctly forced to be stochastic, we specifically set it to transition from $t_{\max}$ to $t_{\max} - 0.0001$, and employ a simple uniformly distributed timestep scheme in $[t_{\min}, t_{\max} - 0.0001)$ for the remaining timesteps, across all settings. For interpolation experiments, to enhance diversity, we increase the step size of the first step from 0.0001 to 0.01.

### F.3 LICENSE

Table 7: The used datasets, codes and their licenses.

| Name | URL | Citation | License |
|---|---|---|---|
| Edges→Handbags | https://github.com/junyanz/pytorch-CycleGAN-and-pix2pix | Isola et al. (2017) | BSD |
| DIODE-Outdoor | https://diode-dataset.org/ | Vasiljevic et al. (2019) | MIT |
| ImageNet | https://www.image-net.org | Deng et al. (2009) | \ |
| Guided-Diffusion | https://github.com/openai/guided-diffusion | Dhariwal & Nichol (2021) | MIT |
| I$^2$SB | https://github.com/NVlabs/I2SB | Liu et al. (2023b) | CC-BY-NC-SA-4.0 |
| DDBM | https://github.com/alexzhou907/DDBM | Zhou et al. (2023) | \ |

We list the used datasets, codes and their licenses in Table 7.

## G ADDITIONAL RESULTS

### G.1 RUNTIME COMPARISON

Table 8 shows the inference time of DBIM and previous methods on a single NVIDIA A100 under different settings. We use `torch.cuda.Event` and `torch.cuda.synchronize` to accurately compute the runtime. We evaluate the runtime on 8 batches (dropping the first batch since it contains extra initializations) and report the mean and std. We can see that the runtime is proportional to NFE. This is because the main computation costs are the serial evaluations of the large neural network, and the calculation of other coefficients requires neglectable costs. Therefore, the speedup for the NFE is approximately the actual speedup of the inference time.

Table 8: Runtime of different methods to generate a single batch (second / batch, $\pm$std) on a single NVIDIA A100, varying the number of function evaluations (NFE).

| Method | NFE | | | |
|---|---|---|---|---|
| | 5 | 10 | 15 | 20 |
| *Center* $128 \times 128$ Inpainting, ImageNet $256 \times 256$ *(batch size = 16)* | | | | |
| I$^2$SB (Liu et al., 2023b) | $2.8128 \pm 0.0111$ | $5.6049 \pm 0.0152$ | $8.3919 \pm 0.0166$ | $11.1494 \pm 0.0259$ |
| DDBM (Zhou et al., 2023) | $2.8711 \pm 0.0318$ | $5.7283 \pm 0.0572$ | $8.3787 \pm 0.1667$ | $11.0678 \pm 0.3061$ |
| DBIM ($\eta = 0$) | $2.8755 \pm 0.0706$ | $5.7810 \pm 0.1494$ | $8.5890 \pm 0.2730$ | $11.1613 \pm 0.3372$ |
| DBIM (2nd-order) | $2.8859 \pm 0.0675$ | $5.7884 \pm 0.1734$ | $8.6284 \pm 0.1907$ | $11.5898 \pm 0.2260$ |
| DBIM (3rd-order) | $2.9234 \pm 0.0361$ | $5.8109 \pm 0.2982$ | $8.6449 \pm 0.2118$ | $11.3710 \pm 0.3237$ |

### G.2 DIVERSITY SCORE

We measure the diversity score (Batzolis et al., 2021; Li et al., 2023) on the ImageNet center inpainting task. We calculate the standard deviation of 5 generated samples (numerical range $0 \sim 255$) given each observation (condition) $x_T$, averaged over all pixels and 1000 conditions.

As shown in Table 9, the diversity score keeps increasing with larger NFE. DBIM ($\eta = 0$) consistently surpasses the flow matching baseline I$^2$SB, and DDBM's hybrid sampler which introduces diversity through SDE steps. Surprisingly, we find that the DBIM $\eta = 0$ case exhibits a larger diversity score than the $\eta = 1$ case. This demonstrates that the booting noise can introduce enough

Table 9: Diversity scores on the ImageNet center inpainting task, varying $\eta$ and the NFE. We exclude statistics for DDBM (NFE$\leq$10) as they correspond to severely degraded nonsense samples.

| Method | | NFE | | | | | | |
|---|---|---|---|---|---|---|---|---|
| | | 5 | 10 | 20 | 50 | 100 | 200 | 500 |
| I$^2$SB (Liu et al., 2023b) | | 3.27 | 4.45 | **5.21** | 5.75 | 5.95 | 6.04 | 6.15 |
| DDBM (Zhou et al., 2023) | | - | - | 2.96 | 4.03 | 4.69 | 5.29 | 5.83 |
| DBIM | $\eta = 0$ | **3.74** | **4.56** | 5.20 | **5.80** | **6.10** | **6.29** | **6.42** |
| | $\eta = 1$ | 2.62 | 3.40 | 4.18 | 5.01 | 5.45 | 5.81 | 6.16 |

stochasticity to ensure diverse generation. Moreover, the $\eta = 0$ case tends to generate sharper images, which may favor the diversity score measured by pixel-level variance.

# H ADDITIONAL SAMPLES

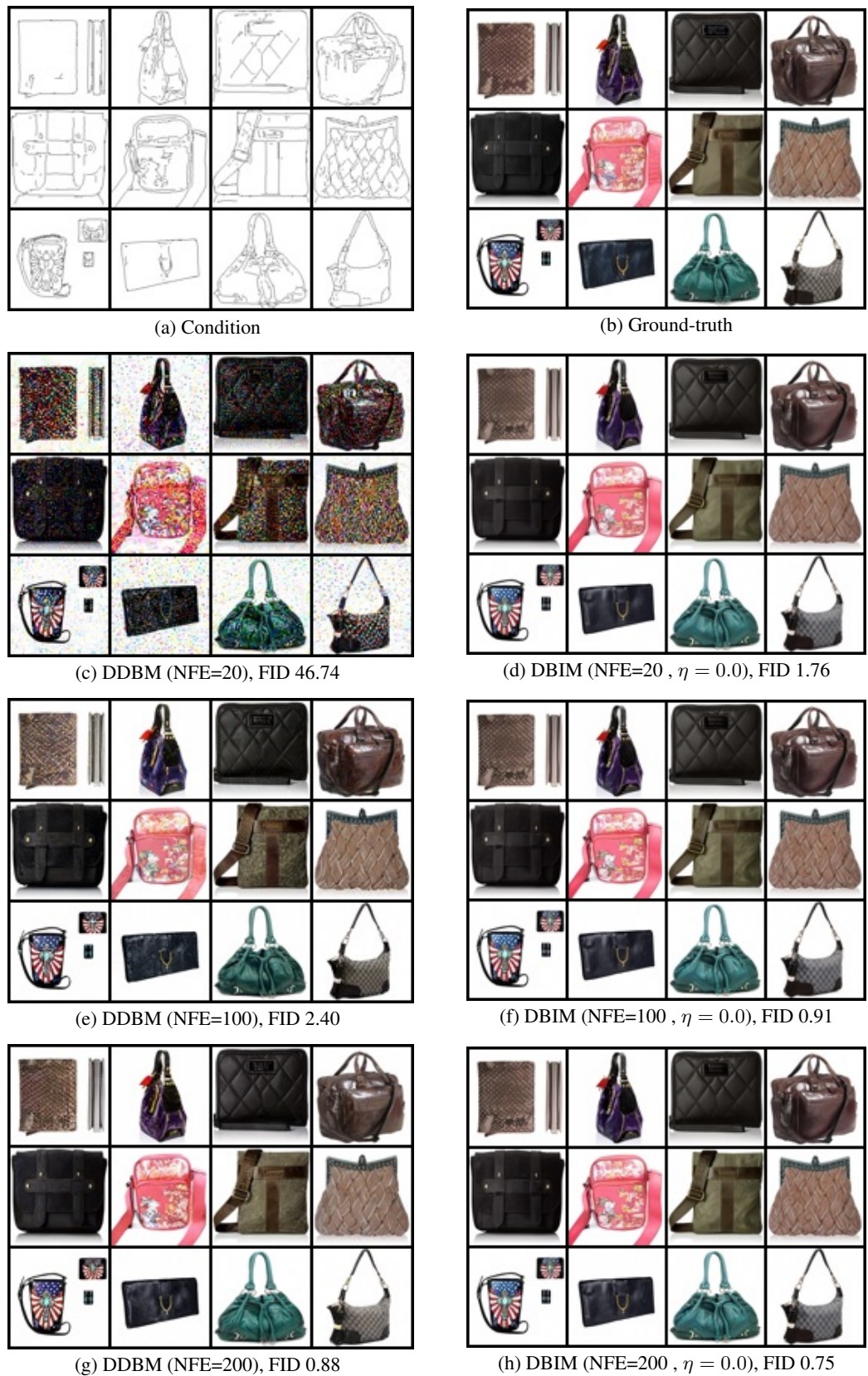

(a) Condition

(b) Ground-truth

(c) DDBM (NFE=20), FID 46.74

(d) DBIM (NFE=20 , $\eta = 0.0$), FID 1.76

(e) DDBM (NFE=100), FID 2.40

(f) DBIM (NFE=100 , $\eta = 0.0$), FID 0.91

(g) DDBM (NFE=200), FID 0.88

(h) DBIM (NFE=200 , $\eta = 0.0$), FID 0.75

Figure 6: Edges→Handbags samples on the translation task.

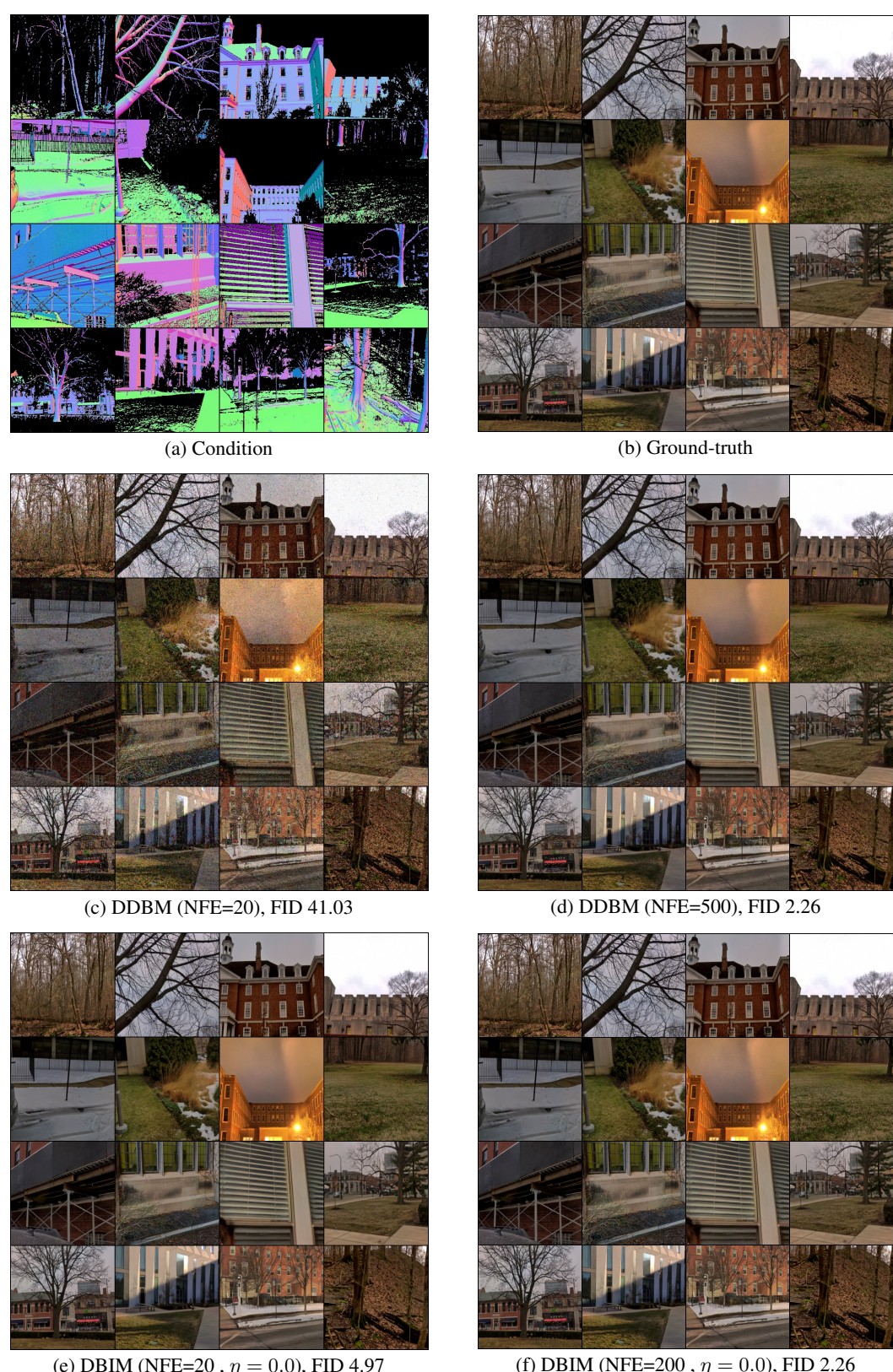

Figure 7: DIODE-Outdoor samples on the translation task.

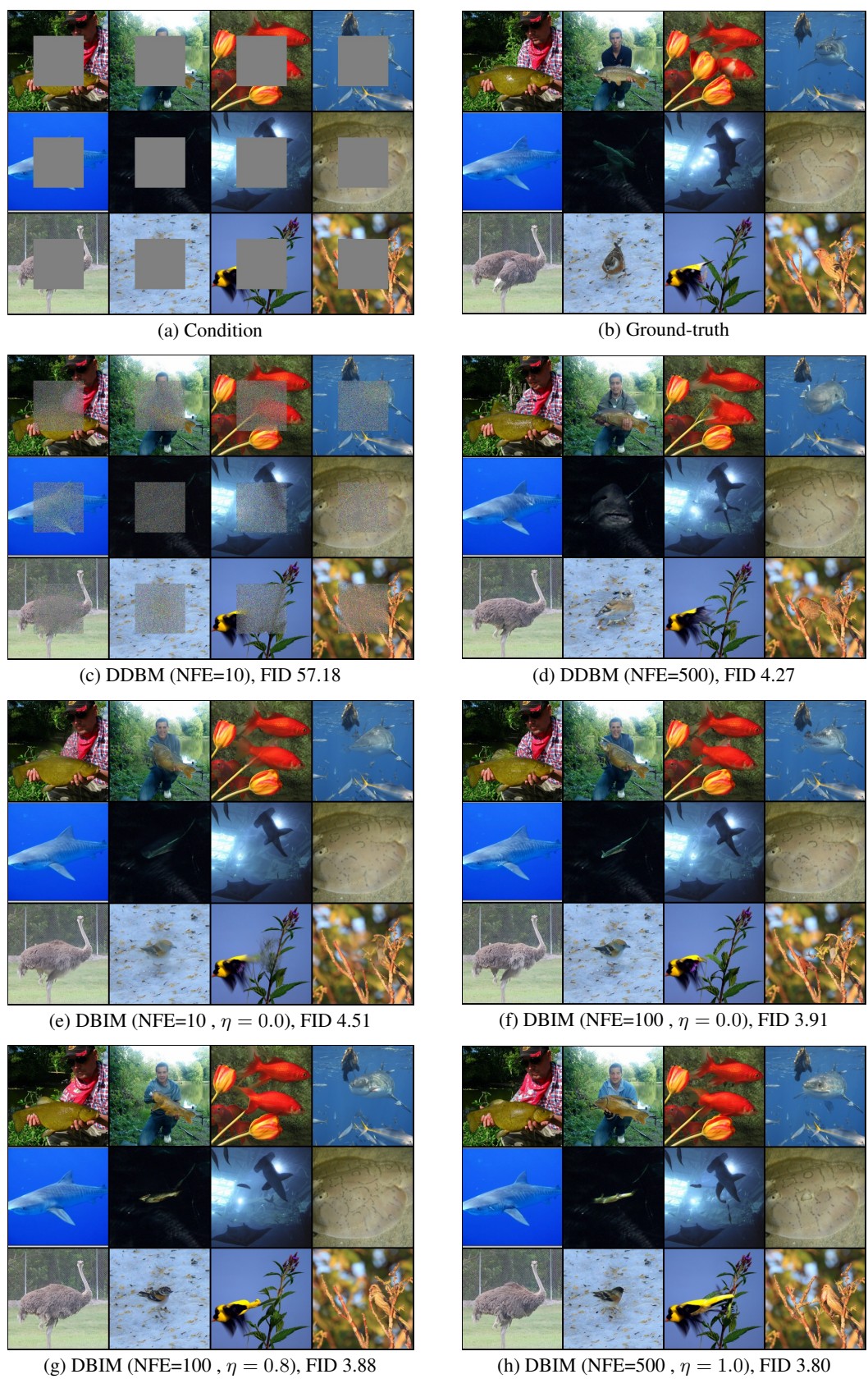

Figure 8: ImageNet $256 \times 256$ samples on the inpainting task with center $128 \times 128$ mask.

