# OpenReview forum: "Diffusion Bridge Implicit Models"
_ICLR.cc/2025/Conference — ICLR 2025 Poster_

### Official Review · Reviewer_YfPj · 2024-10-29

**Soundness:** 3
**Presentation:** 3
**Contribution:** 2
**Rating:** 5
**Confidence:** 4

**Summary:**

The authors present a fast sampling approach to Denoising Diffusion Bridge Models (DDBMs, Zhou et al.) by constructing a DDIM (Song et al.) analogue of DDBMs and hence denote it as Denoising Bridge Implicit Models (DBIMs). The authors empirically demonstrate the advantages of the proposed method on several image translation and restoration benchmarks. The authors also illustrate the application of DBIMs for reconstruction and interpolation using latent space traversal.

**Strengths:**

I found the paper relatively straightforward to follow. Unsurprisingly, the empirical results support the claims in the paper as DDIM leads to significant improvements in sampling efficiency over vanilla DDPM. Therefore, it is to be expected that similar variants of other stochastic processes could achieve similar gains (See [1]).

**Weaknesses:**

Please see below.

**Reinventing the wheel**: My main concern with the paper is that the authors try to reinvent the DDIM framework in the context of a very specific class of stochastic processes (DDBMs in this case) when previous works already define a framework for generalizing over DDIM (as also noted by the authors in the paper in Sec 5.1, higher order methods and Appendix D). More specifically, Pandey et al.  [2] present a generic framework, Conjugate Integrators, for efficient diffusion model sampling of which DDIM and exponential integrators are special cases (See Section 3.1). As a consequence, their framework can be used to enable fast sampling from any pretrained diffusion/flow/interpolant models. Similarly, [1] presents generalized DDIM, which is a special case of Conjugate Integrators and can be applied to diffusion models like CLD or Blurring diffusion models. The main idea of all these methods is to modulate the drift of the reverse diffusion/flow process to enable fast sampling during inference (See also [3] for a similar approach). Since DDBMs also possess a semi-linear drift, extending these generic frameworks to obtain the corresponding DDIM sampler for DDBMs is trivial without delving into the non-markovian formulation of the original DDIM paper.
Consequently, I feel the paper does not offer any new perspective and the flow is very similar to the DDIM paper (defining the non-markovian process followed by setting rho=0 to enable deterministic sampling). Can the authors clarify how their work presents any novel insights compared to the original DDIM paper (or other works mentioned here [1,2]) since this is not obvious to me from the current explanation in the main text?

**Empirical Comparisons**: Although not my main concern, the empirical comparisons seem a bit unfair. For instance, in Table 3, the authors use $\Pi$-GDM and DDRM as comparison baselines, while these methods are task-agnostic and can be adapted for solving different inverse problems using a single pretrained diffusion model prior. However, following Eq. 9, DDBMs/DBIMs require paired data (x_0, y) for training and thus task-specific. Comparing task-specific models with task-agnostic models seems unfair. Moreover, other important baselines based on flow matching [4] or stochastic interpolants [5] are missing, which have advanced quite a lot in terms of sample quality recently [6]. I would recommend updating Table 3 to remove some baselines like DDRM and Pi-GDM and instead compare DBIM with the latest flow matching or interpolant models.

**Dubious claims**: The authors state the following in the Introduction:
```
Adapting diffusion models to scenarios where a more informative prior naturally exists, such as image translation/restoration, requires cumbersome techniques like adding conditions to the model (Ho & Salimans, 2021; Saharia et al., 2022), modifying the generation pipeline (Meng et al., 2022; Su et al., 2022) or adding extra guidance terms during sampling (Chung et al., 2022; Kawar et al., 2022). These approaches inevitably result in either sub-par performance or slow and resource-intensive inference processes.
```
I agree that task-agnostic models like DPS or $\Pi$-GDM require extra guidance terms in their sampling process, making inference slow due to expensive Jacobian-vector products. However, in my opinion, a slower sampling process (though there is some recent work to mitigate slower sampling in solving inverse problems using diffusion models [7,8]) will likely be acceptable over training new models for each task. Moreover, these methods are more scalable in practice due to their adaptiveness to different restoration tasks while requiring no paired data. On the other hand, task-specific methods will likely require large paired datasets, which can be expensive to acquire. Therefore, certain models are better suited for specific applications (as also implied by the No-Free lunch). Therefore, this claim in the introduction seems misleading to me. It would be great if the authors could update this claim to more accurately reflect the nuances of different approaches, including recent advancements in mitigating slower sampling for task-agnostic models. Moreover, it would be nice to discuss how their method fits into the broader context of these trade-offs and under what scenarios it might be preferable to use DBIM versus a more general, task-agnostic method.

References:

[1] gDDIM: Generalized denoising diffusion implicit models, Zhang et al.

[2] Efficient Integrators for Diffusion Generative Models, Pandey et al.

[3] Bespoke Solvers for Generative Flow Models, Shaul et al.

[4] Flow Matching for Generative Modeling, Lipman et al.

[5] Stochastic Interpolants: A Unifying Framework for Flows and Diffusions, Albergo et al.

[6] SiT: Exploring Flow and Diffusion-based Generative Models with Scalable Interpolant Transformers, Ma et al.

[7] Accelerating Diffusion Models for Inverse Problems through Shortcut Sampling, Liu et al.

[8] Fast Samplers for Inverse Problems in Iterative Refinement Models, Pandey et al.

**Questions:**

See above.

---

> ### Author Response · Authors · 2024-11-19
>
> Thank you for your comments. Despite negative, your constructive suggestions are very helpful for improving our work. We provide our responses below.
>
> > Can the authors clarify how their work presents any novel insights compared to the original DDIM paper (or other works mentioned here [1,2]) since this is not obvious to me from the current explanation in the main text?
>
> Thank you for giving us the opportunity to explain where our contribution lies. We are aware of your mentioned works, and we fully understand and agree with your opinion that the underlying idea has been well studied in works like generalized DDIM. For example, we frequently compare with DDIM and related works in Table 1, Appendix C.3/C.4. We believe we have given proper position of our work in the literature and made as many theoretical investigations as we can. We want to argue that, extending these ideas to certain field (diffusion bridge) exhibits differences from prior work, provides new insights, and makes valid contributions to the community.
>
> - **Theoretically**, the generalized DDIM works are distinct from DDBM: (1) though the coefficients before $x_t$ and the network are quite general, even including non-isotropic ones, they don't have the $x_T$ term as DDBM (Equation  (35)) (2) DDBM ends in a delta distribution and has singularity at time $T$, so the first sampling step is forced to be stochastic. However, the forward process considered in generalized DDIMs never ends in delta distribution and never encounters or handles this singularity issue. If DDBM cannot be subsumed by generalized diffusion, then DBIM cannot be subsumed by generalized DDIM.
> - **Technically**, we believe that being simple is important and requires insights. For example, though gDDIM is applicable for general $F$ and $G$, the integral in its Equation 19 only admits closed-form solution in certain scenarios, and deriving these solutions takes effort. If you look at Equation  (35) in our paper, the coefficients are much more complex than the standard diffusion model. Through the DDIM-inspired method, we derive the simplified ODE quite neatly. Another example is that, a key insight of DPM-Solver[2] compared to DEIS[1] is to use the change-of-variable $\lambda_t=\log(\alpha_t/\sigma_t)$ (half of the log-SNR). This greatly simplifies the process of developing high-order solvers, making DPM-solver simpler to implement and easier to follow by subsequent works. We also develop such an insight for DDBMs in our paper: $\lambda_t=\log\left(\frac{b_t}{c_t}\right)=\frac{1}{2}\left(SNR_t-SNR_T\right)$. This is quite different from DPM-Solver, and is novel and insightful to make high-order diffusion bridge solvers work.
> - **Practically**, we have released the code at the anonymous site https://anonymous.4open.science/r/DBIM. We feel that though DDBM has aroused a lot of follow-up works, they all rely on the cumbersome codebase and formulations presented by DDBM. Besides, they are unaware of the possible simplifications and accelerations. Our work definitely provides a fresh perspective to the diffusion bridge community. We believe it can benefit other works based on DDBM, and can inspire future studies. We sincerely hope that our contribution to the community can be recognized. We also note that [3], focusing on fast sampling of a process that is more similar to standard diffusion than us, is well recognized by its reviewers.
>
> [1] Fast sampling of diffusion models with exponential integrator
> [2] DPM-Solver: A Fast ODE Solver for Diffusion Probabilistic Model Sampling in Around 10 Steps
> [3] MRS: A Fast Sampler for Mean Reverting Diffusion based on ODE and SDE Solvers (ICLR2025 Submission, current score 8865)
>
> > I would recommend updating Table 3 to remove some baselines like DDRM and Pi-GDM and instead compare DBIM with the latest flow matching or interpolant models.
>
> Thank you for your suggestion. We have revised the table and use gray color to annotate works that don't require paired training. As for flow matching or interpolant models, we find them mainly focused on image generation instead of image translation tasks, and we don't have their results on our settings (edge2handbags, DIODE, ImageNet inpainting), so we are unable to add them. We have discussed the relationship between DBIM and flow matching in Appendix C.3.

---

> > ### Author Response · Authors · 2024-11-19
> >
> > > It would be great if the authors could update this claim to more accurately reflect the nuances of different approaches, including recent advancements in mitigating slower sampling for task-agnostic models. Moreover, it would be nice to discuss how their method fits into the broader context of these trade-offs and under what scenarios it might be preferable to use DBIM versus a more general, task-agnostic method.
> >
> > Thank you for your suggestion and we totally agree. Actually, we think this point is what DDBM's authors should address, while their paper also only talks about their advantages. We have revised the paper to more accurately reflect the nuances of different approaches.
> >
> > Once again, we are grateful for your constructive feedback and the opportunity to address these issues. We also add proper citations and discussions of your mentioned related works. We hope that the revisions and clarifications we have provided meet your expectations. Please let us know if you have any additional suggestions.

---

> ### Comment · Reviewer_YfPj · 2024-11-25
>
> I want to thank the authors for taking the time to address my concerns and apologies for a delayed response. Here are my thoughts on the author response:
>
> Firstly, the authors point out,
> ```
> Theoretically, the generalized DDIM works are distinct from DDBM: (1) though the coefficients before
>  and the network are quite general, even including non-isotropic ones, they don't have the
>  term as DDBM (Equation (35)) (2) DDBM ends in a delta distribution and has singularity at time
> , so the first sampling step is forced to be stochastic. However, the forward process considered in generalized DDIMs never ends in delta distribution and never encounters or handles this singularity issue. If DDBM cannot be subsumed by generalized diffusion, then DBIM cannot be subsumed by generalized DDIM.
> ```
> I acknowledge the fact that DDBM's have a different structure than gDDIM (of course they do since they are bridges between arbitrary diffusions as compared to DDIM which creates a transport map between Gaussian and data distributions). However, this was not my main concern. As pointed in my review,
> ```
> The main idea of all these methods is to modulate the drift of the reverse diffusion/flow process to enable fast sampling during inference (See also [3] for a similar approach). Since DDBMs also possess a semi-linear drift, extending these generic frameworks to obtain the corresponding DDIM sampler for DDBMs is trivial
> ```
> DDBM like other diffusion/flow models has a semi-linear drift and so it feels trivial to construct samplers like DBIM by existing theory in exponential integrators, conjugate integrators etc.
>
> Secondly, the authors point that,
> ```
> Technically, we believe that being simple is important and requires insights. For example, though gDDIM is applicable for general and, the integral in its Equation 19 only admits closed-form solution in certain scenarios, and deriving these solutions takes effort. If you look at Equation (35) in our paper, the coefficients are much more complex than the standard diffusion model
> ```
> While I agree on the notion of simplicity and acknowledge the contribution in deriving DPM-Solver for bridges, in the worst case implementing gDDIM requires 1 dim numerical integration of coefficient which is trivial so I don't agree here with the latter claim.
>
> Lastly, regarding empirical aspects, the authors state,
> ```
> As for flow matching or interpolant models, we find them mainly focused on image generation instead of image translation tasks, and we don't have their results on our settings (edge2handbags, DIODE, ImageNet inpainting), so we are unable to add them
> ```
> I still feel interpolants are a valid baseline since, in fact, they form a bridge between arbitrary distributions. A lot of recent work in this field has focused on improving interpolants. In light of these improvements, adding such a baseline is important since it informs us when we should use DDBM vs an interpolant.
>
> In light of the theoretical arguments made by the authors, I would increase my rating from 3 to 5; however, I am still leaning towards a rejection since I feel the methodological contributions are marginal and the empirical baselines need to be improved for more clarity on the utility of this method.

---

> > ### Author Response · Authors · 2024-11-25
> >
> > Thank you for acknowledging our efforts and raising the score! We fully understand your comments and decision. We want to present further explanations on why we think our work makes valid contributions to the community. Though we acknowledge that this might not change your opinions on our work, we are still happy to have further discussions.
> >
> > We think on a high level, this is a question of whether "technical" and "practical" efforts [1,2,3,4] can be considered valid contributions. Theoretically, the exponential integrator and gDDIM works tell us that deriving fast samplers for DDBM *is possible* due to the structure of semi-linear drift. In practice, however, it remains to be explored how to implement them, and how they will perform on image translation tasks/DDBMs, instead of image generation tasks/DDPMs. For example, we totally agree with you that
> >
> > > in the worst case implementing gDDIM requires 1 dim numerical integration of coefficient which is trivial so I don't agree here with the latter claim.
> >
> > This is feasible in theory, while in practice there might be concerns about numerical precision (there are singularities at both boundaries in DDBMs, and numerical integration may have larger errors near $t=0$ or $t=T$ than our analytic solution)/additional time cost (which may be small compared to the network evaluation, but better if we can avoid them via our analytic solution). Therefore, we think the "trivial" is only about the theoretical feasibility, instead of technical derivation or practical implementation. If it takes no effort or can be directly done by combining the code of gDDIM and DDBM, we believe the authors of DDBM would have already done so.
> >
> > As for the baseline part, actually, the released checkpoint (as well as the reported results) of I2SB on ImageNet inpainting is quite similar to "flow matching" (they start with noisy conditions instead of clean conditions; they set the intermediate noise level to 0, so no new noise is added). Therefore, we think it can already be seen as a flow matching/interpolant baseline instead of a diffusion bridge model baseline. Also because of that, we train the "real" diffusion bridge model on ImageNet inpainting by ourselves. Note that ImageNet 256x256 is a highly challenging dataset, and **the training takes around 2 weeks on 8xA100**. Therefore, if other papers did not provide results or checkpoints on these tasks, we think we are both not obligated and not resource-possible to help them train models. Even if we do this, we think it may raise concerns about "whether we have thoroughly tuned the hyperparameters/trained for enough long time" for these baselines.
> >
> > Thank you again for your time and efforts in helping us improve our work. We respect your decision and are open to further discussions.
> >
> > [1] Improved techniques for training gans
> > [2] Improved Techniques for Training Score-Based Generative Models
> > [3] Improved Techniques for Training Consistency Models
> > [4] Improved denoising diffusion probabilistic models

---

> > > ### Comment · Reviewer_YfPj · 2024-11-25
> > >
> > > I agree with the sentiment that
> > > ```
> > > "trivial" is only about the theoretical feasibility, instead of technical derivation or practical implementation. If it takes no effort or can be directly done by combining the code of gDDIM and DDBM, we believe the authors of DDBM would have already done so.
> > > ```
> > > and is part of the reason why I decided to increase my score. However, I think that in such cases, where a strong methodological contribution is missing (as also mentioned by Reviewer W5Ra in their Questions section), it is reasonable to expect strong empirical comparisons. In that scenario, flows/interpolants are very strong and obvious baselines that are missing from the revision. I agree that ImageNet 256 can be quite compute intensive and therefore results on other smaller scale datasets would have been a nice addition to the paper. Regardless, I thank the authors for their response and effort into the rebuttal. I will keep my current score.
> > >
> > > P.S. Regarding `This is feasible in theory, while in practice there might be concerns about numerical precision (there are singularities at both boundaries in DDBMs, and numerical integration may have larger errors near
> > >  or
> > >  than our analytic solution)/additional time cost`
> > >
> > > From experience, I want to mention that 1-D numerical integration is very high precision using libraries like Scipy. I have personally not observed any degradation in performance compared to the analytical solution in practice. Regarding additional time cost, the coefficients can be computed offline since they are only dependent on the discretization schedule and thus can be shared between different samples during inference. Therefore, their cost is negligible.

---

> > > > ### Author Response · Authors · 2024-11-25
> > > >
> > > > Thank you again for raising the score and for your great engagement in the discussion process. We mention the numerical issue as we empirically find that computing the coefficients in double-precision floating points performs notably better than single precision, so the algorithm can be sensitive to precision. But we believe in you that libraries like Scipy can handle this well. We also appreciate it if you could consider the I2SB method as a flows/interpolants baseline.

---

### Official Review · Reviewer_W5Ra · 2024-11-02

**Soundness:** 3
**Presentation:** 3
**Contribution:** 2
**Rating:** 6
**Confidence:** 4

**Summary:**

The authors consider the denoising diffusion bridge models (DDBMs [1]) recently proposed as a variant of diffusion models useful for interpolating between two paired distributions. The authors propose a way to accelerate this type of diffusion model without retraining them. The proposed method and the paper are mostly analogical to the Denoising Diffusion Implicit Model (DDIM) [2], which is used to accelerate classical diffusion models for generation such as DDPM [3]. In the experimental section, the authors use the same setups and models as in DDBMs and show that the proposed sampling scheme outperforms the sampling scheme of DDBMs.

**Strengths:**

- The approach is presented and sufficiently supported by the theoretical derivations.
- The authors support their method by showing that it indeed achieves significant speed-up of the inference on the same setups as were considered in the DDBM paper while preserving the quality.
- The method does not require retraining and is implemented just by a different sampling of the DDBMs models.

**Weaknesses:**

The general approach of obtaining a new sampling scheme in analogy to DDIM is known and has already been used for a different type of bridge process in SinSR [4], CVPR 2024, used for super-resolution problems. Specifically, the difference is that in SinSR, the bridge process $q(x_t|x_0, x_T)$ for a given low-resolution image $x_{T} = y$ and a given high-resolution image $x_0$ is given by:
$$q(x_t|x_0, x_T) = \mathcal{N}(x_t|x_0 + \eta_t(y-x_0), k^2\eta_t I), $$
and $\eta_t \rightarrow 1$, while in the current paper, the bridge process $q(x_t|x_0, x_T)$ is given by:
$$
q(x_t \mid x_0, x_T) = \mathcal{N}(a_t x_T + b_t x_0, c_t^2 I),
\quad a_t = \frac{\alpha_t}{\alpha_T} \frac{\text{SNR}_T}{\text{SNR}_t},
\quad b_t = \alpha_t \left( 1 - \frac{\text{SNR}_T}{\text{SNR}_t} \right),
\quad c_t^2 = \sigma_t^2 \left( 1 - \frac{\text{SNR}_T}{\text{SNR}_t} \right),
$$
where $SNR_t = \frac{\alpha_t^2}{\sigma_t^2}$ and $c_t^2 \rightarrow 0$ for $t \rightarrow T$.

Thus, there are two differences. The first one is that in SinSR, the process ends in the Gaussian with center $x_T$, while in the current work, the process ends in the delta function. The second is the different parametrization of these processes. To conclude, the methods SynSR & DBIM (proposed here) are not directly equivalent but one can be derived analogously to the other one with some effort.

**Questions:**

I do not have any particular questions. In my opinion, this is a borderline paper which can be accepted if the space permits.

---

> ### Author Response · Authors · 2024-11-19
>
> Thank you for your positive comments. We provide our responses below.
>
> > To conclude, the methods SinSR & DBIM (proposed here) are not directly equivalent but one can be derived analogously to the other one with some effort.
>
> Thank you for informing us of this work, and we have added it to the related work section. We think it is better to call SinSR a mean-reverting diffusion process, instead of a diffusion bridge process, as we find it similar to [1]. Besides, compared to us, SinSR did not connect it to ODE or develop high-order solvers.
>
> [1] MRS: A Fast Sampler for Mean Reverting Diffusion based on ODE and SDE Solvers (ICLR2025 Submission, current score 8865)

---

> > ### Comment · Reviewer_W5Ra · 2024-11-28
> > **Response**
> >
> > Thanks for you reply. I keep my score.

---

> > > ### Author Response · Authors · 2024-11-28
> > >
> > > Thanks a lot for your consideration and support!

---

### Official Review · Reviewer_Jfwr · 2024-11-03

**Soundness:** 3
**Presentation:** 3
**Contribution:** 3
**Rating:** 6
**Confidence:** 3

**Summary:**

The authors construct an intriguing process that aligns with the marginal distributions and the well-known diffusion bridge. Building on the success of DDBM in modeling paired distributions at given endpoints, the authors introduce Diffusion Bridge Implicit Models (DBIMs), emphasizing discrete steps over continuous diffusion. Robust empirical techniques are employed in real-world experiments, significantly enhancing prediction performance.

**Strengths:**

1. Clear derivation to extend DDBM with DDIM properties.

2. Nice connections to existing literature such as I2SB, flow matching, and exponential integrators.

**Weaknesses:**

the notation of $q_{tT}$ in Eq.(7) is confusing.


NIT - the literature section should be revised significantly, for example, Schrodinger in line 502: Schr\"odinger, schrodinger in line 509, 600, 606, sde line 521, rosenbrock in line 543, Dpm-solver in line 576, ode in line 644 should have properly capitalized letters.

**Questions:**

1. DDBM proposed the pred-x parameterization motivated by the EDM framework to enhance the predictions and derive a set of scaling functions for distribution translation.  Is your work applicable to this extension? If not, what are the challenges?

2. Any intuitions on why a non-Markovian property is preferred?

NIT: some discussions on OT-accelerated diffusions, such as [1,2], may be helpful.

[1] https://arxiv.org/pdf/2110.11291
[2] https://arxiv.org/pdf/2405.04795

---

> ### Author Response · Authors · 2024-11-19
>
> Thank you for your positive comments. We provide our responses below.
>
> > the notation of $q_{tT}$ in Eq.(7) is confusing.
>
> Thank you for pointing out. We have fixed it to $q_{T|t}$.
>
> > NIT - the literature section should be revised significantly, for example, Schrodinger in line 502: Schr"odinger, schrodinger in line 509, 600, 606, sde line 521, rosenbrock in line 543, Dpm-solver in line 576, ode in line 644 should have properly capitalized letters.
>
>
> Thank you for your suggestion. We are unsure about whether we should manually change the titles in the references. We export them directly from Google Scholar. We are afraid that changing them may make the academic indexing systems unable to recognize them.
>
> > DDBM proposed the pred-x parameterization motivated by the EDM framework to enhance the predictions and derive a set of scaling functions for distribution translation. Is your work applicable to this extension? If not, what are the challenges?
>
> Sure, we are already using the pretrained DDBM checkpoints for experiments on edge2handbags and DIODE, which apply complex scaling and skip connections to the network input and output. As detailed in Appendix F.1, as long as we have a "wrapper" that takes xt and outputs x0-prediction, we can substitute it into our algorithm. We have released the code at the anonymous site https://anonymous.4open.science/r/DBIM, and welcome to refer to the `DDBMPreCond` class in `ddbm/karras_diffusion.py`.
>
>
> > Any intuitions on why a non-Markovian property is preferred?
>
> Intuitively, the non-Markovian process reduces the stochasticity in each step (in Equation 11, the variance of the Gaussian is smaller). Using a more deterministic process can reduce the variance of the sampling trajectory and enable faster convergence to the target.
>
> > some discussions on OT-accelerated diffusions, such as [1,2], may be helpful.
>
> Thank you for your suggestions. We have added them to the diffusion Schrodinger bridge part in the related work section.
>
> Once again, thank you for your constructive feedback and for considering our paper for acceptance. We have revised our paper accordingly.

---

> > ### Comment · Reviewer_Jfwr · 2024-11-24
> > **reference**
> >
> > Don't worry about the indexing systems. Consistency and correctness are all you need.

---

> > > ### Author Response · Authors · 2024-11-24
> > >
> > > Thanks for the suggestion again. We agree with you on this point. We have tried to manually change the BibTeX titles to capitalized letters in the source code. However, they still remain lowercase in the paper after compilation. We suspect that the ICLR paper template will force them to be lowercase.

---

### Official Review · Reviewer_hmv5 · 2024-11-04

**Soundness:** 3
**Presentation:** 2
**Contribution:** 3
**Rating:** 6
**Confidence:** 3

**Summary:**

The paper proposes an extension of DDBM by adopting the ideas from DDIM for the bridge problem (translating one distribution to another with diffusion).

**Strengths:**

The paper novel in the sense that it proposes to adopt the ideas from DDIM (fast sampling, non-markovian diffusion) to the bridge problem, thereby enhancing DDBMs and expanding the practitioner’s toolset for such kind of problems. So, “creative combinations of existing ideas” - is the best description of the presented research. Also, I would like to notice good experimental section.

**Weaknesses:**

The majority of my concerns are in the "Questions" section. To be honest, I didn’t check Proposition 3.2 - but it seems to be correct. Proposition 4.1 - also not carefully checked by me, but seems to be correct.
What I do not like much about the paper is that some particular formulas and results (see point 2 in my questions) are stated as is, without proof/reference to the proof. It complicates reading and checking the results.
Some other weaknesses:
1. You have "Related works" section (A) in the Appendix. Ok, but I didn't find a reference to this section from the main text. Furthermore, I think it is important to somehow cover related works in the main text (maybe, with some references to the appendix)
2. No source code in the submission. It is important for reproducibility.

**Questions:**

1. Appendix C.1, lines 892-897. Why do you take gradient $\nabla_{x_{t_{n+1}}} \log q^{\rho} (x_{t_{n + 1}} \vert x_0, x_T)$, not $\nabla_{x_{t_{n+1}}} \log q^{\rho} (x_{t_{n + 1}} \vert x_0, x_{t_n}, x_T)$. Also, the eq. 50 is incorrect, while the result (lines 904 - 906) is correct. Please check the derivation.
2. Line 256 - The statement that $x_T$ is cancelled out under the choice of $\rho_n$ seems to be correct, but a link to the proof should be given.
3. Do I understand correctly, that different variants of variance parameter $\rho$ appear only at inference, i.e., training with eq. 13 is agnostic to $\rho$?
4. You mention that eq. (8) (from DDBM) is equivalent to (16). But on par with (16) you have additional booting noise required by theory. How previous works (e.g., DDBM) manage to avoid the necessity of this noise?
5. Lines 427 - 428: ‘straightforward mapping without the involvement of stocahsticity’ - but you have booting noise still, yes?

---

> ### Author Response · Authors · 2024-11-19
>
> Thank you for your positive comments. We appreciate that you carefully read the math and proofs. We provide our responses below.
>
> > I didn't find a reference to this section from the main text. Furthermore, I think it is important to somehow cover related works in the main text (maybe, with some references to the appendix)
>
> Thank you for your suggestion. We have added a small related work section in the main text with references to the appendix.
>
> > No source code in the submission. It is important for reproducibility.
>
> We agree. We have released the code at the anonymous site https://anonymous.4open.science/r/DBIM to ensure reproducibility.
>
>
> > Appendix C.1, lines 892-897. Why do you take gradient $\nabla_{x_{t_{n+1}}} \log q^{\rho} (x_{t_{n + 1}} \vert x_0, x_T)$, not $\nabla_{x_{t_{n+1}}} \log q^{\rho} (x_{t_{n + 1}} \vert x_0, x_{t_n}, x_T)$. Also, the eq. 50 is incorrect, while the result (lines 904 - 906) is correct. Please check the derivation.
>
> Thanks for pointing out! We do miss $x_{t_n}$ in several places and have fixed them.
>
>
> > Line 256 - The statement that $x_T$ is cancelled out under the choice of $\rho_n$ seems to be correct, but a link to the proof should be given.
>
> Thank you for your suggestion. We briefly prove it in the footnote.
>
> > Do I understand correctly, that different variants of variance parameter $\rho$ appear only at inference, i.e., training with eq. 13 is agnostic to $\rho$?
>
> Yes, as $\rho$ only affects the relative loss weightings at different timesteps for DBIM. It does not affect the global minimum, which is minimum at all timesteps. Appendix C.2 also gives explanation.
>
> > You mention that eq. (8) (from DDBM) is equivalent to (16). But on par with (16) you have additional booting noise required by theory. How previous works (e.g., DDBM) manage to avoid the necessity of this noise?
>
> DDBM divides each step into 2 substeps: (1) simulating SDE and (2) simulating ODE. Therefore, the first step is always simulating the SDE, which has no singularity. $\eta=1$ in DBIM can be seen as a better discretization of the SDE than the Euler step.
>
> > Lines 427 - 428: ‘straightforward mapping without the involvement of stocahsticity’ - but you have booting noise still, yes?
>
> Yes, our sampling always has the booting noise. We are expressing that maybe directly mapping the condition is enough. But in practice, the first step needs stochasticity to avoid the singularity. Only stochasticity in intermediate timesteps can be omitted.
>
> Once again, thank you for your constructive feedback and for considering our paper for acceptance. We have revised our paper accordingly.

---

> > ### Comment · Reviewer_hmv5 · 2024-11-25
> > **Thank you**
> >
> > I thank the authors for the answer, my questions are clarified. In light of the other reviews, I think my score is good enough.

---

> > > ### Author Response · Authors · 2024-11-25
> > >
> > > Thank you for your positive feedback! We are glad that our responses help.

---

### Official Review · Reviewer_Wtri · 2024-11-08

**Soundness:** 3
**Presentation:** 3
**Contribution:** 3
**Rating:** 8
**Confidence:** 3

**Summary:**

The paper presents a theoretically justified method for fast sampling from DDBMs. The authors introduce non-Markovian diffusion bridges and derive the sampling procedure with the same marginals for discrete time steps. They show that the training objective of the proposed methods is equivalent to the DDBM objective on the discretized timesteps. The method is validated on image-to-image translation problems using edges2handbags and DIODE-Outdoor datasets and inpainting tasks using the ImageNet dataset. The proposed method achieves comparable results with DDBM on NFE > 100 while using only NFE = 20 for these problems regarding FID, LPIPS, and MSE.

**Strengths:**

1. The paper proposes fast sampling for the DDBM method, which achieves impressive results for image-to-image translation problems. Enhancing and accelerating DDBM is essential since these models provide a solid alternative to Flow Matching and Schrodinger Bridge methods for image-to-image problems.
2. The proposed method is training-free and doesn't require any model parameter fine-tuning. Proposition 3.2 establishes equivalence between DDBM and DBIM objectives up to the constant.
3. Experimental results, which prove that deterministic sampler with $\eta = 0$ leads to better results than stochastic samplers with $\eta > 0$ is novel for DDBM, which suggests operating with stochastic samplers and may provide "averaged" or blurry outputs given initial conditions. This result allows further improvement of DDBM with distillation methods for deterministic samplers like consistency distillation (Consistency Models, ICML-2023).

**Weaknesses:**

In my opinion, the paper doesn't have significant weaknesses. However, some comments should be discussed additionally:
1. I think comparing the best competitor on the inpainting problem according to Table 3 - the I2SB method - is not enough. According to the I2SB paper, this method can repaint the masked region with semantic structures with only 2-10 NFEs for the inpainting problem with freeform masks. I suggest that the authors compare their method with I2SB for the inpainting problem with freeform masks and show the dependence of its quality (LPIPS, FID) on small NFEs (1-20) following setup from I2SB.
2. The idea of incorporating non-Markovian diffusion bridges seems not novel. For example, the I3SB method (Implicit Image-to-Image Schrodinger Bridge for Image Restoration, https://arxiv.org/abs/2403.06069) shares similar ideas and parameterization for I2SB. I suggest the author add the discussion of this paper and the main differences between their method and I3SB.
3. The diversity of the proposed method should be evaluated quantitatively. It is worth showing how small $\eta$ and NFE values degrade diversity compared to stochastic models. I suggest that the authors compute the diversity score from CMDE (Conditional Image Generation with Score-Based Diffusion Models, https://arxiv.org/abs/2111.13606) as in BBDM (BBDM: Image-to-image Translation with Brownian Bridge Diffusion Models, CVPR-2023).

**Questions:**

Questions:
1. Can you compare your method with I2SB for inpainting with freeform masks and evaluate I2SB on small NFEs?
2. Can you discuss the limitations of your method or provide failure cases?
3. Can you comment on how much slower high-order methods are than first-order methods regarding inference time? Does the quality improve further for bigger orders like 4?
4. Can you additionally provide inference time for DBIM and other methods to show the inference time acceleration?
5. I suggest the authors consider the other method of DDBM acceleration using consistency distillation and include its discussion in the literature review; see Consistency Diffusion Bridge Models (NeurIPS-2024).

Typos:
1. It looks like in Eq. 7, it should be \bar{w}_{t} instead of w_{t}.

---

> ### Author Response · Authors · 2024-11-19
>
> Thank you for your positive comments. We provide our responses below.
>
> > The diversity of the proposed method should be evaluated quantitatively.
>
> Thank you for your suggestion. On the ImageNet inpainting task, following CMDE and BBDM, we calculate the standard deviation of 5 generated sampled (numerical range 0~255) given each observation (condition), averaged over all pixels and 1000 conditions.
>
> |NFE|5|10|20|50|100|200|500|
> |:--------|:-------:|:-------:|:------:|:------:|:-------:|:-------:|:------:|
> |$\eta=0$|**3.74**|**4.56**|**5.20**|**5.80**|**6.10**|**6.29**|**6.42**|
> |$\eta=1$|2.62|3.40|4.18|5.01|5.45|5.81|6.16|
>
> The diversity score keeps increasing with more NFE. Surprisingly, we find that the our $\eta=0$ case exhibits larger diversity score than the $\eta=1$ case. This demonstrates that the booting noise can introduce enough stochasticity to ensure diverse generation. Moreover, the $\eta=0$ case tends to generate sharper images. This may favor the diversity score which is measured by pixel-space variance.
>
> > Can you compare your method with I2SB for inpainting with freeform masks and evaluate I2SB on small NFEs?
>
> Our inpainting model on ImageNet is trained by ourselves, not directly taken from I2SB. The reason is that, the checkpoints I2SB released are actually not the standard bridge as DDBM: for inpainting, they start from condition with randomly added noise, instead of clean condition; starting from the noisy condition, they set the intermediate noise level as 0, so no more noise is additionally added, and the model becomes an interpolant (or flow matching) model.
>
> As the training takes around 2 weeks on 8xA100 GPUs, we are unable to additionally training the free-form inpainting model in the rebuttal period. Center inpainting is a more difficult task than freeform inpainting (as in I2SB paper, its FID is higher), so we believe experiments on center inpainting is convincing. We report the FIDs of I2SB (center inpaint) under small NFEs as follows:
>
> |NFE|5|10|15|20|
> |:--------|:-------:|:-------:|:------:|:------:|
> |I2SB|6.43|5.24|5.07|4.98|
> |DBIM ($\eta=0$)|**6.11**|**4.52**|**4.24**|**4.15**|
>
> We currently compute the coefficients $a_t,b_t,c_t$ in 64-bit to enhance precision, so the DBIM result is slightly better than the original paper. I2SB also performs reasonably under small NFE, as its posterior sampling is the same as DBIM($\eta=1$) under its noise schedule (Appendix C.3).
>
> > Can you discuss the limitations of your method or provide failure cases?
>
> Sure. Please refer to the conclusion section in the revised paper.
>
> > Can you comment on how much slower high-order methods are than first-order methods regarding inference time? Does the quality improve further for bigger orders like 4? Can you additionally provide inference time for DBIM and other methods to show the inference time acceleration?
>
> We report the average sampling time (in seconds) of 8 batches on a single A100, under the batch size of 16.
>
> |NFE|5|10|15|20|
> |:--------|:-------:|:-------:|:------:|:------:|
> |I2SB|2.8128 $\pm$ 0.0111|5.6049 $\pm$ 0.0152|8.3919 $\pm$ 0.0166|11.1494 $\pm$ 0.0259|
> |DDBM|2.8711 $\pm$ 0.0318|5.7283 $\pm$ 0.0572|8.3787 $\pm$ 0.1667|11.0678 $\pm$ 0.3061|
> |DBIM ($\eta=0$)|2.8755 $\pm$ 0.0706|5.7810 $\pm$ 0.1494|8.5890 $\pm$ 0.2730|11.1613 $\pm$ 0.3372|
> |DDBM (2nd-order)|2.8859 $\pm$ 0.0675|5.7884 $\pm$ 0.1734|8.6284 $\pm$ 0.1907|11.5898 $\pm$ 0.2260|
> |DDBM (3rd-order)|2.9234 $\pm$ 0.0361|5.8109 $\pm$ 0.2982|8.6449 $\pm$ 0.2118|11.3710 $\pm$ 0.3237|
>
> As the networks are of the same architecture and size, the inference time is approximately proportional to the NFE. Other costs, like computing the coefficients in the sampler, are negligble. Higher orders than 3 will not further improve the quality. With higher orders, the estimation error of the higher-order derivatives will also increase. As seen in Table 6, the improvement from 2nd-order to 3rd-order is already small and even worse. According to the experiences in diffusion model solvers, order 3 or even 2 is the best.
>
>
> > Discuss some missing related work
>
> Thanks for pointing out! We have added proper citations and discussions in the related work section.
>
>
> > Typo: It looks like in Eq. 7, it should be $\bar w_t$ instead of $w_t$.
>
> Thanks for careful reading! We have fixed it in the revised paper.
>
> Once again, thank you for your constructive feedback and for considering our paper for acceptance. We have revised our paper accordingly. We have also released the code at the anonymous site https://anonymous.4open.science/r/DBIM to ensure reproducibility.

---

> > ### Comment · Reviewer_Wtri · 2024-11-26
> > **Thank you**
> >
> > I thank the authors for their reply, which clarified my questions. The only comment on the diversity scores is that I suggest the authors to show the diversity scores not only for their methods but also for the methods of their competitors. The reason is to show that the proposed model can generate diverse samples as well as competing methods. I will save my score.

---

> > > ### Author Response · Authors · 2024-11-26
> > >
> > > We are glad that our responses help clarify your concerns! We appreciate your positive feedback and your constructive suggestions for improving our work. We'll make our supplementary experiments more complete in the final version of the paper.

---

### Meta-Review · Area_Chair_KMoE · 2024-12-19

**Metareview:**

This paper proposes a fast sampling method for Denoising Diffusion Bridge Models (DDBMs) which can interpolate two arbitrary paired distributions. Inspired by diffusion bridges, a non-Markovian process is introduced which requires knowledge of $x_0$ at inference time, which is then approximated with its mean. Empirically improved performance was demonstrated. Although some of reviewers have unresolved concerns about novelty and significance, overall the strengths overweigh weaknesses in my opinion, and I'm pleased to recommend acceptance.

**Additional Comments On Reviewer Discussion:**

Although some of reviewers have unresolved concerns about novelty and significance, overall the strengths overweigh weaknesses in my opinion.

---

### Decision · Program_Chairs · 2025-01-22

Accept (Poster)